# Parameterization of the collision-coalescence process using series of basis functions: *COLNETv1.0.0* model development using a machine learning approach

Camilo Fernando Rodríguez-Genó[1], Léster Alfonso[2]

[1]Atmospheric Sciences Centre, National Autonomous University of Mexico, Mexico City, 04510, Mexico
[2]Autonomous University of Mexico City, Mexico City, 09790, Mexico

*Correspondence to*: Camilo Fernando Rodríguez-Genó (camilo.rodriguez@atmosfera.unam.mx)

**Abstract.** A parameterization for the collision-coalescence process is presented, based on the methodology of basis functions. The whole drop spectrum is depicted as a linear combination of two lognormal distribution functions, leaving no parameters fixed. This basis-function parameterization avoids the classification of drops in artificial categories such as cloud water (cloud droplets) or rain water (raindrops). The total moment tendencies are predicted using a Machine Learning approach, in which one deep neural network was trained for each of the total moment orders involved. The neural networks were trained and validated using randomly generated data, over a wide range of parameters employed by the parameterization. An analysis of the predicted total moment errors was performed, aimed to establish the accuracy of the parameterization at reproducing the integrated distribution moments representative of physical variables. The applied machine learning approach shows a good accuracy level when compared to the output of an explicit collision-coalescence model.

**Keywords:** cloud microphysics; collision-coalescence; lognormal distribution; microphysics parameterization; numerical modelling; machine learning; neural networks.

## 1 Introduction

Drop populations are represented using drop size distributions (DSD). The first attempt at characterizing drop spectra was made by Marshall and Palmer (1948), who employed exponential distributions based on drop diameter to describe the DSDs. More recently, the use of a three-parameter gamma distribution has shown a good agreement with observations (Ulbrich, 1983). However, lognormal distributions have shown a better squared-error fit to measurements of rain DSDs than gamma or exponential distributions (Feingold and Levin, 1986; Pruppacher and Klett, 2010). The analysis of several important characteristics of the Brownian coagulation process showed that the lognormal distribution adequately represents the particle distributions (Lee et al., 1984, 1997). In addition, some authors have employed this type of distribution function, lognormal, to parameterize cloud processes with promising results (Clark, 1976; Feingold et al., 1998; Huang, 2014).

There are two main approaches to modelling cloud processes: the explicit approach (bin microphysics) and the bulk approach
(bulk microphysics). Bin microphysics is based on the discretization of a DSD into sections (bins), and calculates the evolution
of the DSD due to the influence of different processes that could be dynamical and/or microphysical (Berry, 1967; Berry and
Reinhardt, 1974; Bott, 1998a; Khain et al., 2004, 2010). The core of this method is the solution of the Kinetic Coagulation
Equation (KCE) (von Smoluchowski, 1916b, 1916a) for the collision-coalescence of liquid drops, (also known as Stochastic
Coalescence Equation or Kinetic Collection Equation within the cloud physics community), in a previously designed grid,
which could be over mass or radius. Thus, previous knowledge of the characteristics or parameters of the distributions is not
necessary. This way of solving the KCE is very accurate, but its operational utility is compromised because it is
computationally very expensive, due to the need to calculate a large number of equations, ranging from several dozens to
hundreds, at each grid point and time step. Besides, as the KCE has no analytical solution, it has to be solved via numerical
schemes, which are very diffusive by nature. While diffusive schemes could be appropriate for certain microphysical processes
such as sedimentation (Khain et al., 2015), it is a disadvantage that has to be dealt with. However, the numerical solutions of
the KCE have evolved in such a way that today we can find  models that are specifically designed to limit the diffusiveness of
these numerical methods (Bott, 1998a).

In the case of bulk microphysics, the KCE is parameterized and the evolution of a chosen set of statistical moments related to
physical variables is calculated, instead of the evolution of the DSD itself. A pioneer approach to this kind of parameterizations
can be found in Kessler (1969), where a simple but relatively accurate representation of the autoconversion process is
introduced. One or two-moment parameterizations are common (Cohard and Pinty, 2000; Lim and Hong, 2010; Milbrandt and
McTaggart-Cowan, 2010; Morrison et al., 2009; Thompson et al., 2008). However, recently it has been extended to three-
moment parameterizations (Huang, 2014; Milbrandt and Yau, 2005). This type of parameterization is computationally
efficient, which makes it popular within the operational weather forecasting community. The main disadvantage of this
approach is that the equations for solving the rates of the $p$-th moment include moments of a higher order, so the system of
equations employed to calculate the evolution of the moments is not closed (Seifert and Beheng, 2001). This could be avoided
by using predefined parameters for the distributions that describe the DSD, which normally take the form of exponential
(Marshall and Palmer, 1948), gamma (Milbrandt and McTaggart-Cowan, 2010; Milbrandt and Yau, 2005) or lognormal
distributions (Huang, 2014). Besides, artificial categories are often used to separate hydrometeors (cloud and rain water),
depending on drop radius, values between 20 $\mu m$ and 41 $\mu m$ are examples of employed threshold values (Cohard and Pinty,
2000; Khairoutdinov and Kogan, 2000), with the moments for each category being calculated by means of partial integration
of the KCE.

An additional approach to modelling microphysical processes is the particle-based one, which is based on the application of a
stochastic model such as the Monte Carlo method to the coagulation (coalescence) of drop particles inside a cloud. This method
has been approached from a number of perspectives. For example Alfonso et al. (2008) analysed the possible ways of solving
the KCE by using a Monte Carlo algorithm and several collision kernels, with good correspondence between the analytical
and numerical approaches for all the kernels, by estimating the KCE following Gillespie's Monte Carlo algorithm (Gillespie,

1972) and several analytical solutions. Also, the possible implications of this approach for cloud physics are discussed. Other variants of this approach are analysed in Alfonso et al. (2011), and it has also been used to simulate the subprocesses of autoconversion and accretion applying a Monte Carlo-based algorithm within the framework of Lagrangian cloud models (Noh et al., 2018). This approach is accurate, and represents well the stochastic nature of the collision-coalescence of drops, but it is also computationally expensive, as a large number of particles are needed in each grid cell, to be able to calculate accurate statistics (Morrison et al., 2020). The cost of these schemes could be reduced by using simple methods to treat droplet activation, such as the Twomey CNN activation (Grabowski et al., 2018; Twomey, 1959). However, even considering those simplifications, the cost of a Lagrangian particle-based scheme is 25% greater than bin microphysics, when considering a similar number of particles and bin variables per grid cell (Grabowski, 2020; Morrison et al., 2020).An alternative to these main approaches is a hybrid approach to parameterize the cloud microphysical processes. This approach simulates the explicit approach in the way that it describes the shape of the DSD through a linear combination of basis functions (Clark, 1976; Clark and Hall, 1983), and it could be considered a middle point between bulk and bin microphysics. This is done by having time-varying distribution parameters, instead of fixed ones, as is common in conventional bulk approaches. A system of prognostic equations is solved to obtain the parameters' tendencies related to the statistical distribution functions based on the evolution of their total moments (the combination of the statistical moments with same order of all distribution functions involved), describing their tendencies due to condensation and collision-coalescence. Since the integration process covers the entire size spectrum, the artificial separation of the droplet spectrum is avoided, making the terms cloud droplet and rain drop meaningless (they are just drops), and it is possible to solve a fully closed system of equations without the need to keep any parameter of the distribution constant. However, this integration can be made only once for all parameters at each time step. Another advantage of this approach is its independence from a specific collision kernel type, as is common in the bulk approach; in order to obtain analytical expressions from the integrals of the KCE, a polynomial type kernel such as the one derived by Long (1974) is frequently used. Having said that, a limitation of this approach is that the total moment tendencies have to be solved at each time step for the needed parameters. An alternative solution for this shortcoming is previously calculating the moment's rates by including a sufficiently wide range of parameters, and store the results in lookup tables that should be consulted several times at every time step.

Machine Learning (ML) is the study of computer algorithms that improve automatically through experience and by the use of data (training) (Mitchell, 1997). ML algorithms build a model based on sample data in order to make predictions or decisions without being explicitly programmed to do so (Koza et al., 1996). They are used in a wide variety of applications, such as in medicine, email filtering, and computer vision, where it is difficult or unfeasible to develop conventional algorithms to perform the needed tasks. In particular, neural networks (NN) are especially well suited for solving non-linear fitting problems and for establishing relationships within complex data such as the outputs of the KCE. In the field of atmospheric sciences, the use of Deep Neural Networks (DNN) has been extended to the parameterization of sub-grid processes in climate models (Brenowitz and Bretherton, 2018; Rasp et al., 2018), while in cloud microphysics, the autoconversion process was parameterized using DNNs with a superior level of accuracy when compared with equivalent bulk models (Alfonso and Zamora, 2021; Loft et al.,

2018). Also, a partial parameterization of collision-coalescence was tested in Seifert and Rasp (2020), which developed a ML parameterization that includes the processes of autoconversion and accretion, describing the droplet spectra as a gamma distribution, and establishing a comparative study that exposed the advantages and disadvantages of the use of ML techniques on cloud microphysics.

In order to eliminate the need to solve the rate equations for the total moments of the KCE at every time step (Thompson, 1968), or resort to the use of lookup tables, we propose to predict the total moment tendencies using a ML approach within this parameterization. Thus, the objective of this study is to apply DNN to the parameterized formulation of the collision-coalescence process developed by Clark (1976) in order to replicate the rate equations for the total moments, eliminating the need of lookup tables or numerical solution of integrals. By doing this, a complete representation of collision-coalescence, based on ML, could be achieved (except drop breakup).

The research article is structured as follows: In section 2, the parameterization framework is described, as well as the reference model used for comparison purposes; In section 3, the procedures of DNN methodology are explained and the network architecture is introduced, the training data set is generated, and the DNN is trained and validated; In section 4, the experiment design is explained; In section 5, the results of the experiment are discussed, an assessment of the results is made by contrasting them with the reference solution, and the predicted total moment errors are analyzed; and in section 6 several conclusions are drawn.

## 2 Description of the collision-coalescence parameterization

### 2.1 Formulation of the total moment tendencies

Under the framework of the parameterization developed in this study, any given drop spectrum can be approximated by a series of basis functions. Therefore, the distribution that characterizes the evolution of the spectrum is given by a linear combination of probability density functions as shown below:

$$f\langle r \rangle = \sum_{i=1}^{I} f_i\langle r \rangle \qquad (1)$$

where $f_i\langle r \rangle$ are the individual members of the set of distributions considered, $I$ stands for the number of distribution functions that make up the set, and $r$ refers to the radius of drops. In the case at hand, a set of two statistical distributions is employed. At each time step, the rates of the parameters of each distribution will be calculated. It should be noted that, in any set of distributions considered, all the members have the same type of distribution. For the proposed parameterization, as described in Clark (1976), a distribution of log-normal type is used, as follows

$$f\langle r \rangle = \frac{N}{\sqrt{2\pi}\sigma r} e^{[-(\ln r - \mu)^2/(2\sigma^2)]} \qquad (2)$$

Where μ and $\sigma^2$ stand for the mean and variance of $\ln r$ respectively, while $N$ represents the number concentration of drops. Considering that moment of order $p$ $(\overline{R^p})$ of any distribution can be defined as (Straka, 2009)

$$N\overline{R^p} = \int_0^\infty r^p f(r) dr \qquad (3)$$

the following analytical solution of eq. (3) can be found for the moments of the lognormal distribution

$$\overline{R^p} = e^{p\mu + \frac{1}{2}p^2\sigma^2} \qquad (4)$$

Combining eqs. (1), (3) and (4), the $p$-th total moment of a linear combination of lognormal distributions could be expressed as (Clark and Hall, 1983)

$$N\overline{R^p} = \sum_{i=1}^I N_i \overline{R_i^p} = \sum_{i=1}^I N_i e^{p\mu_i + \frac{1}{2}p^2\sigma_i^2} \qquad (5)$$

Where the index $i$ indicates each of the individual members of the set ($I=2$). Deriving eq. (5) with respect to time, we obtain the tendencies of the total moments of a series of log-normal distributions

$$\frac{\partial N\overline{R^p}}{\partial t} = \sum_{i=1}^I N_i \overline{R_i^p} \left( \frac{\partial \ln N_i}{\partial t} + p \frac{\partial \mu_i}{\partial t} + \frac{p^2}{2} \frac{\partial \sigma_i^2}{\partial t} \right) \qquad (6)$$

Equation (6) can be expressed as a system of equations

$$AX = F \qquad (7)$$

where $\mathbf{X}$ is a vector representing the tendencies of the distribution parameters

$$X^T = \left[ \frac{\partial \ln N_1}{\partial t}, \frac{\partial \ln N_2}{\partial t}, \dots, \frac{\partial \ln N_I}{\partial t}, \frac{\partial \mu_1}{\partial t}, \frac{\partial \mu_2}{\partial t}, \dots, \frac{\partial \mu_I}{\partial t}, \frac{\partial \sigma^2_1}{\partial t}, \frac{\partial \sigma^2_2}{\partial t}, \dots, \frac{\partial \sigma^2_I}{\partial t} \right] \qquad (8)$$

The coefficient's matrix $\mathbf{A}$ is a squared matrix of order $v$ ($v = 3 \times I$) defined as

$$A = \begin{cases} a_{i,j} = N_j \overline{R_j^{i-1}} / (N\overline{R^{i-1}}) \\ a_{i,j+I} = (i-1)N_j \overline{R_j^{i-1}} / (N\overline{R^{i-1}}) \\ a_{i,j+2I} = \frac{1}{2}(i-1)^2 N_j \overline{R_j^{i-1}} / (N\overline{R^{i-1}}) \end{cases} \qquad (9)$$

with $i = 1,2,\dots,v$ and $j = 1,2,\dots,I$. The components of the independent vector $\mathbf{F}$ are the tendencies of the total moments of the distributions:

$$F^T = \left[ \frac{\partial \ln N\overline{R^0}}{\partial t}, \frac{\partial \ln N\overline{R^1}}{\partial t}, \dots, \frac{\partial \ln N\overline{R^{v-1}}}{\partial t} \right] \qquad (10)$$

Both $\mathbf{A}$ and $\mathbf{F}$ are normalized in order to achieve a better numerical stability in the solution of the system of equations. The evolution of the distribution functions' parameters is calculated by applying a simple forward finite differences scheme (Clark and Hall, 1983)

$$N_i^{k+1} = N_i^k e^{\frac{\partial \ln N_i^k}{\partial t} \Delta t} \qquad (11a)$$

$$\mu_i^{k+1} = \mu_i^k + \frac{\partial \mu_i^k}{\partial t} \Delta t \qquad (11b)$$

$$(\sigma^2)_i^{k+1} = (\sigma^2)_i^k + \frac{\partial (\sigma^2)_i^k}{\partial t} \Delta t \qquad (11c)$$

With $k$ being the time index in the finite difference's notation.

## 2.2 Description of the calculation of the total moment tendencies due to collision-coalescence

The KCE determines the evolution of a DSD due to collision-coalescence. This equation can be expressed in a continuous form as a function of mass as follows (Pruppacher and Klett, 2010)

$$\frac{\partial f}{\partial t} = \int_0^{m/2} f(m - m')f(m')K(m - m'|m')dm' - \int_0^\infty f(m)f(m')K(m|m')dm' \qquad (12)$$

where $K(m|m')$ is the collection kernel. Reformulating eq. (12) in the form of Thompson (1968)and in function of radius, we can calculate the total moment tendencies (vector **F** from the previous section) as follows

$$\frac{dN\overline{R^p}}{dt} = \frac{1}{2} \int_0^\infty \int_0^\infty B^p(r_1, r_2)K(r_1|r_2)f\langle r_1 \rangle f\langle r_2 \rangle dr_1 dr_2 \qquad (13)$$

where

$$B^p(r_1, r_2) = (r_1^3 + r_2^3)^{p/3} - r_1^p - r_2^p \qquad (14)$$

$$K\langle r_1|r_2 \rangle = \pi(r_1 + r_2)^2 E(r_1, r_2)|V_T(r_1) - V_T(r_2)| \qquad (15)$$

Equation (15) represents the hydrodynamic kernel and $E(r_1, r_2)$ stands for the collection efficiencies taken from Hall (1980), which is based on a lookup table representing the effectiveness of drop collisions under given environmental conditions. A set of two lognormal distributions (eq. (2)) is used as members of the set in eq. (1). Hence, the prognostic variables under the parameterization formulation will be the corresponding parameters of both distribution functions: $N_1$, $\mu_1$ , $\sigma_1$, $N_2$, $\mu_2$ and $\sigma_2$. At this point in the parameterization the total moment tendencies should be calculated either by solving eq. (13) at each time step for all the moments involved, or by searching in a lookup table calculated a priori. Instead, the following chapter explains in detail the ML approach proposed and its implementation.

**2.3 Description of the reference model**

To obtain a reference solution (KCE from now onwards), the explicit model developed by Bott (1998a, 1998b) was employed. This scheme is conservative with respect to mass and very efficient computationally speaking. It is based on the numerical integration of the KCE (eq. 12), combined with the mass density function $g(y,t)$ (Berry, 1967), in order to simplify the calculations:

$$g(y,t)dy = mn(x,t)dx \qquad (16)$$

$$n(m,t) = \frac{1}{3m^2}g(y,t) \qquad (17)$$

where $y = \ln r$ and $r$ is the radius of a drop of mass $m$. By substituting (17) in (12) we obtain the KCE for the mass density function (Bott, 1998a)

$$\frac{dg(y,t)}{dt} = \int_{y_0}^{y_1} \frac{m^2}{m_c^2 m'}g(y_c,t)K(y_c,y')g(y',t)dy' - \int_{y_0}^{\infty} g(y,t)\frac{K(y,y')}{m'}g(y',t)dy' \quad (18)$$

where $m_c = m - m'$. The first integral of the right-hand side of eq. (18) represents the gain of drops of mass $m$ due to collision-
coalescence of two smaller droplets, while the second integral portrays the loss of drops of mass $m$ being captured by bigger drops (Bott, 1998a). For the numerical solution of eq. (18), a logarithmic equidistant mass grid is used, and is generated as

$$m_{k+1} = \alpha m_k, \qquad k = 1,2,\dots,n \qquad (19)$$

where $n$ is the total number of grid points.

**3 Machine Learning architecture and training data set**

ML methodology can be classified into three main categories, according to the problem at hand: supervised, unsupervised and reinforced learning. In our case, supervised learning is used. Supervised learning algorithms build a mathematical model of a set of data that contains both the inputs and the desired outputs (Russell and Norvig, 2010). Under this classification, there is previous knowledge of the set of input values $\vec{x}_k$, and their corresponding outputs $\vec{y}_k$, , with $k = 1,2,\dots,n$, where n is the amount of input values. The objective is to obtain a function $f(\vec{x})$, by means of which the new data $\vec{x}_{new}$ simulates reasonably
well the output values. The set $\{\vec{x}_k, \vec{y}_k\}$; $k = 1,2,\dots,n$ is called the training data set. To test the performance of $f(\vec{x})$, the input and output data are separated into two different data sets: training and testing. As NN are able to fit any non-linear function (Schmidhuber, 2015), a ML parameterization should approximate reasonably well the solution of the KCE in the form of eq. (13), given enough layers and neurons in the architecture of the network.

### 3.1 Neural network architecture

Deep Neural Networks are based on artificial neurons. Each neuron receives a set of input data, processes it and passes it to an activation function $\sigma(z)$, which returns the activated output (Fig. 1). The activation value of neuron $i$ in layer $l$ is denoted by $a_i^l$ and is determined as

$$a_i^l = \sigma\left(z_i^l\right) \tag{20}$$

$$z_i^l = b_i^l + \sum_{j=!}^{m_l-1} W_{i,j}^l x_i^{l-1} \tag{21}$$

In eq. (21), $b_i^l$ is the bias, $W_{i,j}^l$ is the ponderation weight, $m_{l-1}$ the number of neurons in layer $l$-$1$, $\sigma(z)$ is the activation function, and $z$ is the processing intermediate value of the variable. Hence, a NN could be defined as a set of input values ($\vec{x}$), bias values ($\vec{b}$) and weights ($\overrightarrow{W}$) integrated in a functional form, i.e. $\vec{y}(\vec{x}, \overrightarrow{W}, \vec{b})$, and its training procedure consists of minimizing an error function (known as loss function), by optimizing the weights and biases for the available training data. A commonly used loss function is the regression mean squared error (MSE). Hence, we need a minimization algorithm to process the following expression

$$C(\overrightarrow{W}, \vec{b}) = \frac{1}{2n}\sum_k \left\|\vec{y}(\vec{x}_k, \overrightarrow{W}, \vec{b}) - \vec{y}_k\right\|^2 \tag{22}$$

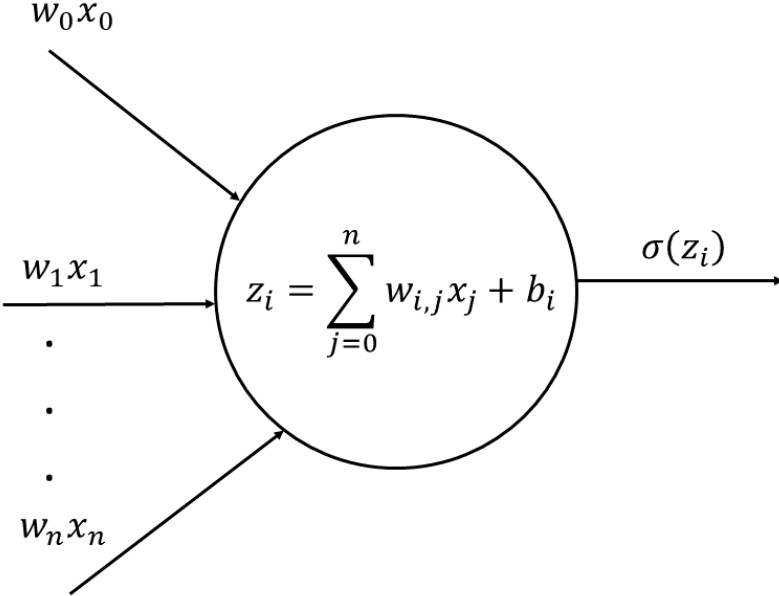

**Figure 1: Schematic representation of an artificial neuron.**

The selected algorithm for minimization of the loss function (eq. (22)) is the bayesian regularization, which updates the weight

and bias values according to the Levenberg-Marquardt optimization (Marquardt, 1963). Backpropagation is used to calculate the Jacobian of the performance with respect to the weight and bias variables (Dan Foresee and Hagan, 1997; MacKay, 1992). The designed DNN is conformed by one layer which receives the input data (input layer), three intermediate layers (hidden layers) with 20 neurons each and an output layer with a single neuron which returns the simulated target values (Fig. 2).

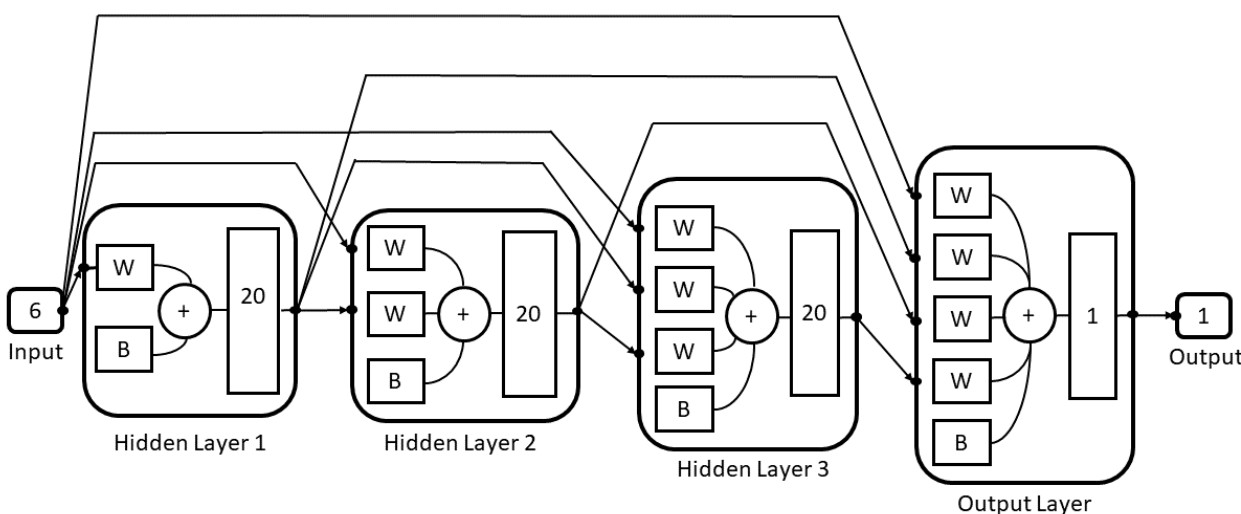

**Figure 2: Schematic representation of the architecture of the trained neural network used to calculate the total moment tendencies. The neural network receives six inputs and then processes them by means of three hidden layers of 20 neurons each, and an output layer with a single neuron and one output value.**

**3.2 Generation of the training and validation data sets**

The training procedure consists of feeding the DNN with six input values corresponding to the distribution parameters of each

distribution and the total moment tendency for the $p$-th order obtained from eq. (13) as a target. The NN training algorithm then processes those values in order to establish the relationships between the data provided. This process is repeated until all input and target data are processed. The resulting trained DNN should be able to estimate the total moment tendencies from a given set of distribution parameters that falls within the ranges of the training variables. A schematic representation of the trained NN with the inputs and output is shown in Fig. 3.

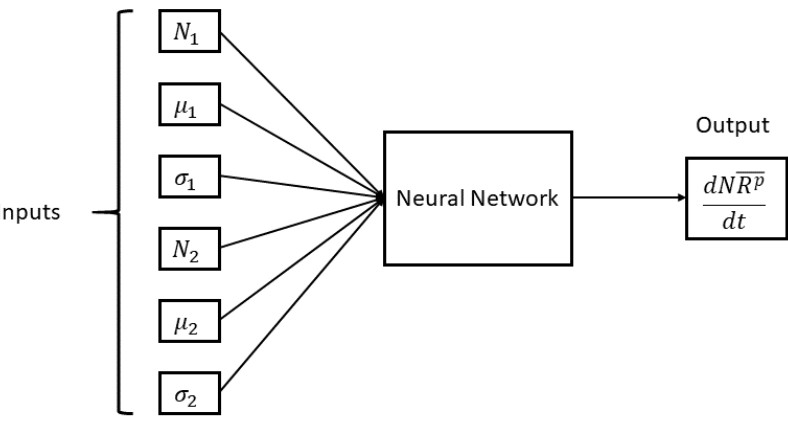

**Figure 3: Neural network parameterization inputs and output. The input data are the six distribution parameters $(N_1, \mu_1, \sigma_1, N_2, \mu_2$ and $\sigma_2)$ needed to feed eq. (13), while the output is the $p$-th order total moment tendency $\left(\frac{dN\overline{R^p}}{dt}\right)$.**

In order to generate the training and test data sets, 100000 drop spectra derived from the input variables are employed, over a wide range of distribution parameters $(N_1, \mu_1, \sigma_1, N_2, \mu_2$ and $\sigma_2)$ . Those input parameters are used to calculate the total moment rates from eq. (13) and train the DNN. Five DNNs are trained, one for each total moment tendency involved in the formulation of the parameterization (moment orders ranged from 0 to 5), with exception of the total moment of order 3, as total mass is not affected by the collision-coalescence process. The same training input parameters are used to train all NNs, varying only the target values corresponding to the total moment tendencies of each order.

The physical variables related to the input parameters are shown in Fig. 4 for a better representation of the generated training clouds. The training and test data is created using an uniformly distributed random number generator, with means and standard deviations shown in Table 1, as well as the ranges (minimum and maximum values) of each predictor variable.

**Table 1: Statistical description of the input values used in the training and test data sets. The means, standard deviation and ranges are shown for each input variable.**

| Input Variable | Mean | Standard Deviation | Range [min, max] |
|---|---|---|---|
| Concentration (N) $(cm^{-3})$ | 250.80 | 144.13 | [1.00; 500.00] |
| μ Parameter | -7.00 | 0.58 | [-8.00; -6.00] |
| σ Parameter | 0.20 | 0.06 | [0.10; 0.30] |

Figure 4 shows that within the ranges of the training data (concentration from 1 cm$^{-3}$ to 500 cm$^{-3}$), the corresponding liquid water contents (LWC) are between $10^{-10}$ g cm$^{-3}$ and $10^{-4}$ g cm$^{-3}$, with the majority of the data concentrated between the limits

of $10^{-8}$ g cm$^{-3}$ and $10^{-5}$ g cm$^{-3}$. Those values cover a sufficiently wide range of liquid water content to adequately represent warm clouds within the parameterization.

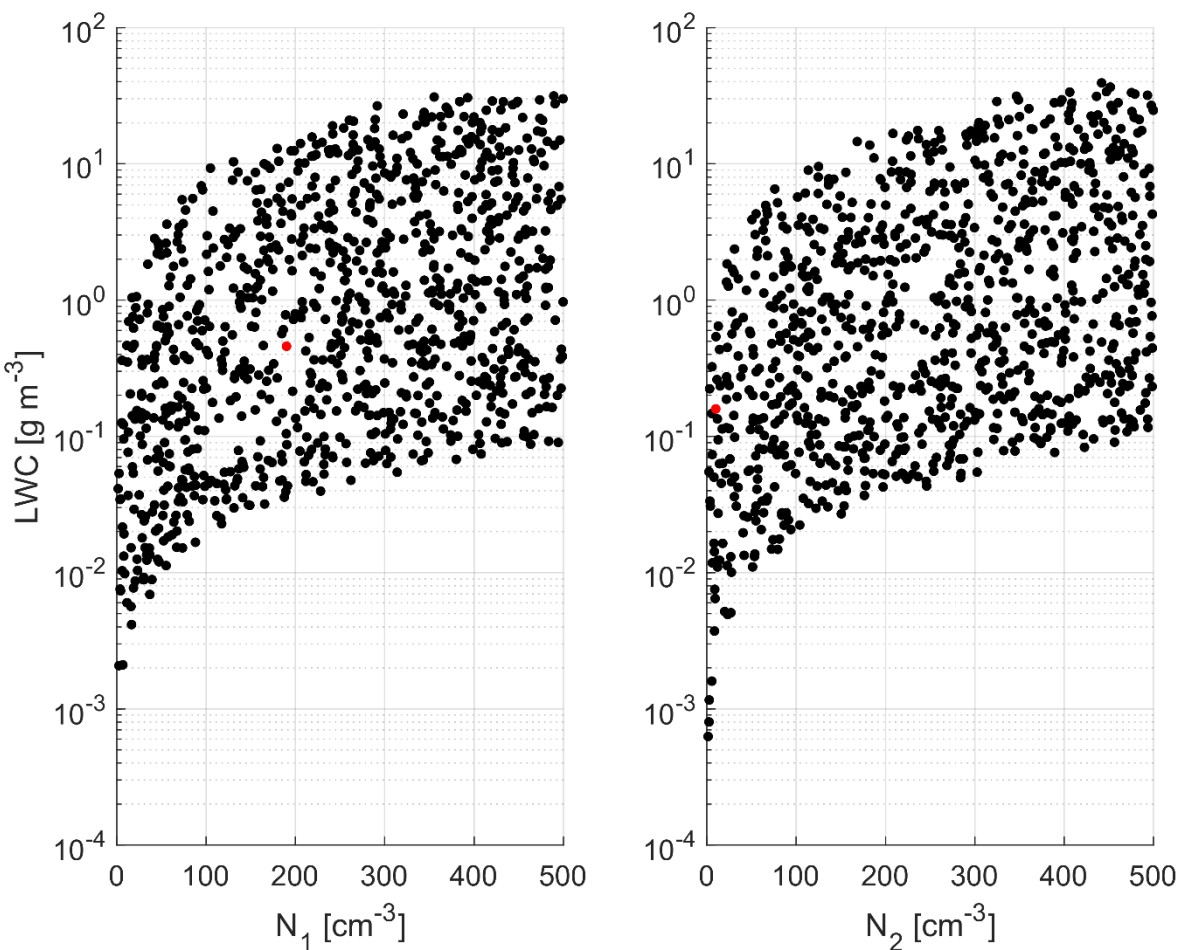

**Figure 4: Scatterplot of liquid water content (LWC) calculated from the input parameters of $f_1$ (left) and $f_2$ (right) vs drop number concentration. The LWC values are obtained from the statistical moment of order 3 using the parameters depicted in Table 1, and were calculated from eq. (4). The red dots represent the initial conditions for the experiment case included in Table 4. Only one in hundred data points are shown.**

### 3.3 Training and testing of the Deep Neural Network

From the available data, 80 % is employed in training the DNN, and the remaining 20 % is used for testing purposes. The total moment tendencies (eq. 13) are solved using a trapezoidal rule, over a logarithmic radius grid between the ranges of $1\ \mu m \leq r \leq 10^4\ \mu m$. The solutions of eq. (13) are called the target values. The mean and standard deviation for each calculated total moment rate are shown in Table 2.

**Table 2: Means and standard deviations of total moment tendencies (target values) for each statistical moment used. The data is calculated from eq. (13) with the distribution parameters ($N_1, \mu_1, \sigma_1, N_2, \mu_2$ and $\sigma_2$) as input values.**

| Total Moment Order | Mean | Standard Deviation |
|---|---|---|
| M0 | -0.0021 | 0.0014 |
| M1 | -0.0015 | 0.0011 |
| M2 | -0.0009 | 0.0006 |
| M4 | 0.0011 | 0.0007 |
| M5 | 0.0024 | 0.0016 |

Both input and target values are normalized as follows

$$x_{norm} = \frac{x - \bar{x}}{\sigma} \qquad (23)$$

The input and target values require normalization to facilitate the work of the optimization algorithm. All nodes in each layer of the DNN use the MSE as a loss function. The training procedure for a NN consists of processing a fragment of the total training data through the network learning architecture, then determining the prognostic error and the gradient of the loss

function (MSE) back through the network in order to update the weight values. This algorithm is repeated via an iterative process over all training data until the performance index (MSE) is small enough or a predefined number of passes through all data are completed. One pass through all training data is known as an epoch. In this case, a maximum number of 1000 epochs is established, and a minimum value of $10^{-7}$ is considered for the gradient function.

Five DNN are trained, one for each total moment tendency involved in the formulation of the parameterization (moment orders

ranged from 0 to 5, except 3$^{rd}$ moment). A variant of the training process, known as cascade-forward neural network training, is employed. This enables the network to establish more precise non-linear connections between the data provided, but results in a more computationally expensive training process (Karaca, 2016; Warsito et al., 2018). The main difference with the standard training procedure (feed-forward) is that it includes a connection from the input and every previous layer to following layers (see Fig. 2). As with feed-forward networks, a two-or more layer cascade network can learn any finite input-target

relationship arbitrarily well, given enough hidden neurons. The total moment tendencies for the statistical moment of order 3 is not calculated because the collision-coalescence process does not affect total mass.

Performance (MSE) training records for the total moment tendencies calculated from eq. (13) are depicted in Fig. 5. The speed of convergence is similar in all cases, and all networks converged at epoch 1000. This occurs because the gradient value never was below the minimum, so the training process kept refining the results until it reached the maximum number of epochs

previously defined. Despite that, a good performance is achieved, with the MSE in the order of $10^{-4}$ for all orders of the total moment tendencies as shown in Table 3, where the best (final) MSE values for each trained DNN are manifested in detail. Since the values of the total moments are normalized in the DNN model (scale of $10^0$), these values of MSE are considered as

the indications of high accuracies for the scale of the problem. Thus, the NN model developed reproduces very well the rates of the moments following the formulation of Clark (1976), which is a main objective in this research

**Table 3: Best performance in the training process of the DNNs. The performance measures are the Mean Squared Error (MSE) and the Pearson Correlation Index. The shown data correspond to the total moment tendencies obtained from the trained neural networks, with input values and reference targets taken from the validation data set.**

| Total Moment Order | Best Performance (MSE) | Correlation Index |
|:---:|:---:|:---:|
| M0 | 2.59e-04 | 0.9998 |
| M1 | 3.49e-04 | 0.9998 |
| M2 | 2.68e-04 | 0.9999 |
| M4 | 1.80e-04 | 0.9999 |
| M5 | 2.05e-04 | 0.9998 |

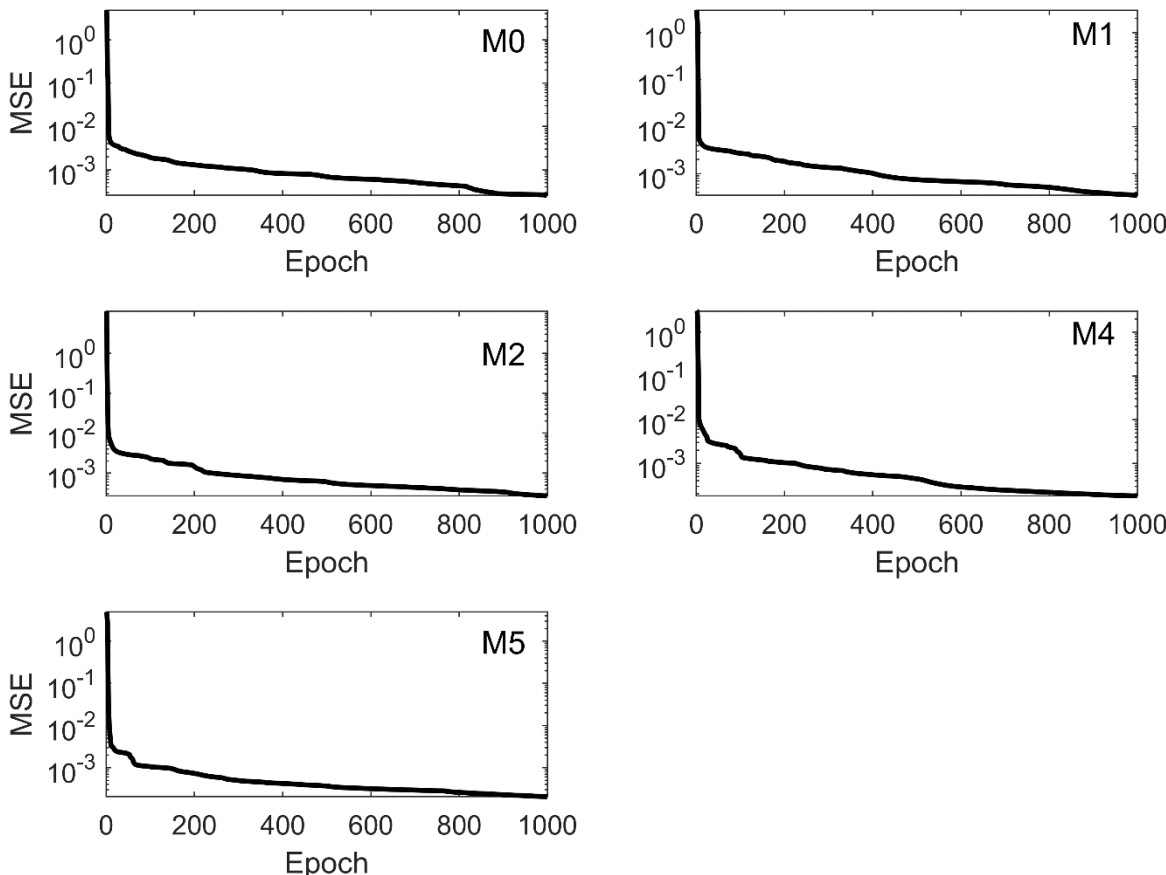

**Figure 5: Performance training records of total moment tendencies for the moments from order 0 to 5 (3rd order is excluded). The shown data corresponds to the total moment tendencies obtained from the trained neural networks, with input values and reference targets taken from the validation data set. The performance measure is the Mean Square Error (MSE).**

Regression plots for the trained networks are depicted in Fig. 6. It is a comparison between the outputs obtained from evaluating the trained NNs using the test inputs and the targets from the test data set corresponding to each of the total moment tendencies obtained from eq. (13). Minor differences can be appreciated from the graphics, with the trained DNN models overestimating or underestimating the actual values. However, a good agreement was reached for all trained DNN, with the predicted values from the DNN matching the actual output from the solution of eq. (13) with a coefficient of correlation between 0.9998 and 0.9999 in all cases (as shown in Table 3). The axis ranges of the graphics vary because the plotted data is non-normalized, thus, there are different ranges for each of the calculated total moment tendencies. This result means that the rates of the total

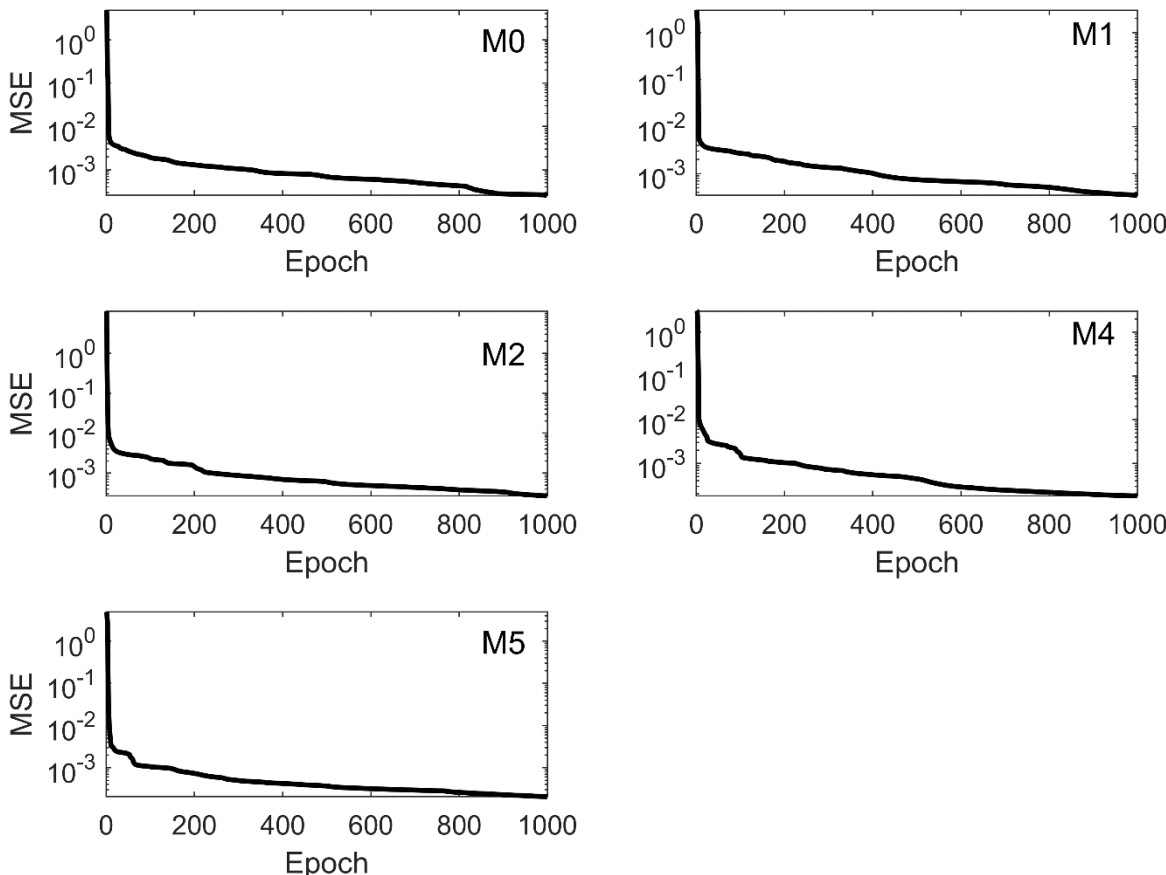

moments obtained from Clark's parameterization of collision-coalescence (Clark, 1976), are accurately reproduced by the NN model developed.

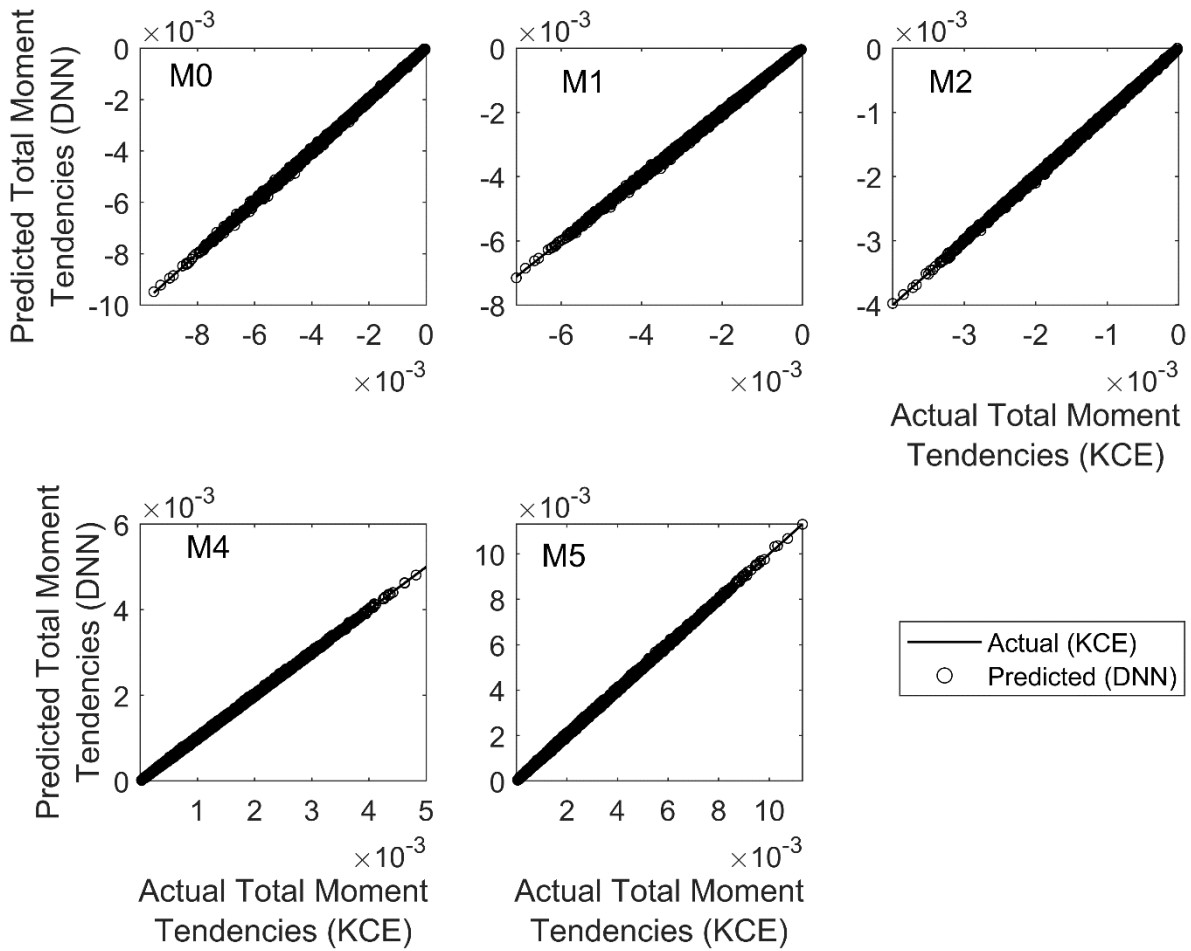

**Figure 6: Regression plots for the five DNN trained. It is a comparison between the outputs obtained from evaluating the trained neural networks using the test inputs and the targets from the validation data set corresponding to each of the total moment tendencies obtained from eq. (13). The *y* axis varies for each subplot because the plotted data is non-normalized.**

Experiments with non-normalized training data were performed, yielding results with MSE at least an order of magnitude higher. Those results are not shown in the present article due to the lower accuracy of the regression outputs.

Neural networks give us a better way to estimate the values of the integral (13).. The neural networks of course do not replace the computation of integrals, but since they have the ability to learn and model complex non-linear functions, they allow (once trained) to estimate them efficiently for values of the parameters ($N_1$, $\mu_1$, $\sigma_1$, $N_2$, $\mu_2$ and $\sigma_2$), for which it has not been previously calculated.

Before the widespread adoption of machine learning, the alternative previously used by other authors (Clark, 1976; Clark and Hall, 1983; Feingold et al., 1998) were the lookup tables, that are tables that stores a list of predefined values (the moment tendencies in this case). Then, in the context of our work, the lookup table is a mapping function that relates the parameters of the basis functions ($N_1$, $\mu_1$, $\sigma_1$, $N_2$, $\mu_2$ and $\sigma_2$), with the total moment tendencies $\left(\frac{dN\overline{R^p}}{dt}\right)$.

However, usually, functions computed from lookup tables have a limited domain.. Preferably, we need functions whose domain is a set with contiguous values. Furthermore, every time we need to calculate the integral (13), a search algorithm must be executed in order to retrieve the moment tendency for a given set of parameters, and some kind of interpolation will be needed to compute moment tendencies for values of the parameters for which it has not been calculated.

The advantage of the neural networks is that all the computational effort is dedicated to the training phase. Once we trained the networks, they replace the lookup tables and are able to map the parameters of the basis functions with total moment tendencies.

## 4 WDM6 parameterization and experiment design

An experiment is performed with the objective of illustrating the behaviour of the ML-based parameterized model (P-DNN) and how it compares with the results of a traditional bulk parameterization and the reference model (KCE). This experiment should not be interpreted as an evaluation of the overall behaviour of P-DNN, but as an example of how it predicts the DSD and bulk variables. A detailed evaluation of the novel components of the P-DNN scheme was already carried out in the previous chapter.

### 4.1 Initial conditions and experiment design

A simulation covering $t = 900\,s$ (15 minutes) of system evolution is considered for all models, with a time step of $\Delta t = 0.1\,s$. The initial parameters for the distribution functions of the parameterized model are as shown in Table 4.

**Table 4: Initial parameters for the distribution functions of P-DNN. Each distribution is characterized by a concentration parameter ($N$), expected value ($\mu$) and standard deviation ($\sigma$). The initial parameters are shown for the two lognormal distribution functions employed in the formulation of P-DNN.**

| Parameter | $f_1$ | $f_2$ |
|---|---|---|
| N | 190 cm$^{-3}$ | 10 cm$^{-3}$ |
| μ | -7.1505 | -6.5219 |
| σ | 0.1980 | 0.1980 |

The values from Table 4 are well within the parameters established on Table 1, and were set following Clark (1976). The initial spectrum for the KCE was calculated from these parameters to ensure the same initial conditions for both models. A 300-point logarithmic equidistant grid was generated for the integration of the KCE, with radii values in the range of

$0.25 \ \mu m \leq r \leq 2.6 \times 10^4 \ \mu m$. Equations (16) and (17) were used to transform the output of both models to make them comparable, while the bulk quantities from the KCE were integrated from the calculated spectra.

## 4.2 WDM6 parameterization

To better establish the accuracy of the P-DNN, an extra parameterization was included in the comparison with the reference solution. The selected parameterization is the WRF Double Moment 6-class bulk mode (WDM6) (Lim and Hong, 2010), which was chosen for being implemented in a well-known three-dimensional atmospheric model (WRF). The collision-coalescence section of that parameterization is explained in detail in Cohard and Pinty (2000), and treats the main warm microphysical processes in the context of a bulk two-moment framework. A scheme of such a type is believed to be a pragmatic compromise between bulk parametrizations of precipitation as proposed by Kessler (1969) and very detailed bin models. Inclusion of a prognostic equation for the number concentration of raindrops provides a better insight into the growth of large drops, which in turn can only improve the time evolution of the mixing ratios.

The scheme makes use of analytical solutions of the KCE for accretion and self-collection processes, while autoconversion follows the formulation of Berry and Reinhardt (1974). This has been done by using the generalized gamma distribution, which enables fine tuning of the DSD shape through the adjustment of two dispersion parameters. All the tendencies, except the autoconversion of the cloud droplets, are parametrized based on continuous integrals that encompass the whole range of drop diameters within each water substance category (cloud droplets and raindrops). The threshold for considering a change of category is $r = 41 \ \mu m$. Thus, the gamma distributions employed are truncated in this radius value. With this method, the treatment of autoconversion is the weakest link in the scheme because this process acts in the diameter range where the fuzzy transition between cloud droplets and raindrops is hardly compliant with a bimodal and spectrally wide (from zero to infinity) representation of the drops. As neither the KCE model (Bott, 1998a), the Clark's parameterization (Clark, 1976) or the developed ML model take into account drop breakup, the formulation of this process included in Cohard and Pinty (2000) has been left out of the current implementation. This model is referred to as P-CP2000 in the following sections. For comparison purposes, all simulations share the same initial conditions. It should be noted that WDM6, being a conventional two-moment scheme, is focused on the evolution of the moments of order zero and three of a truncated gamma distribution function.

## 5 Discussion of results

The results shown in this section were obtained using the parameterized model COLNETv1.0.0 (Rodríguez-Genó and Alfonso, 2021a, 2021c, 2021b).

## 5.1 Spectra comparison

The output of this parameterized Deep Neural Network model (P-DNN) are the updated distribution parameters at every time step ($N_1$, $\mu_1$, $\sigma_1$, $N_2$, $\mu_2$ and $\sigma_2$). The physical variables related to the moments of the distributions, such as mean radius or

liquid water content (LWC) are diagnosed from those parameters. Besides, we can calculate the shape and scale of the drop spectrum at any given time, by integrating the distribution functions defined by its parameters.

Figure 7 shows a comparison between the mass density spectra derived from P-DNN and KCE models for three chosen times (300 s, 600 s and 900 s).

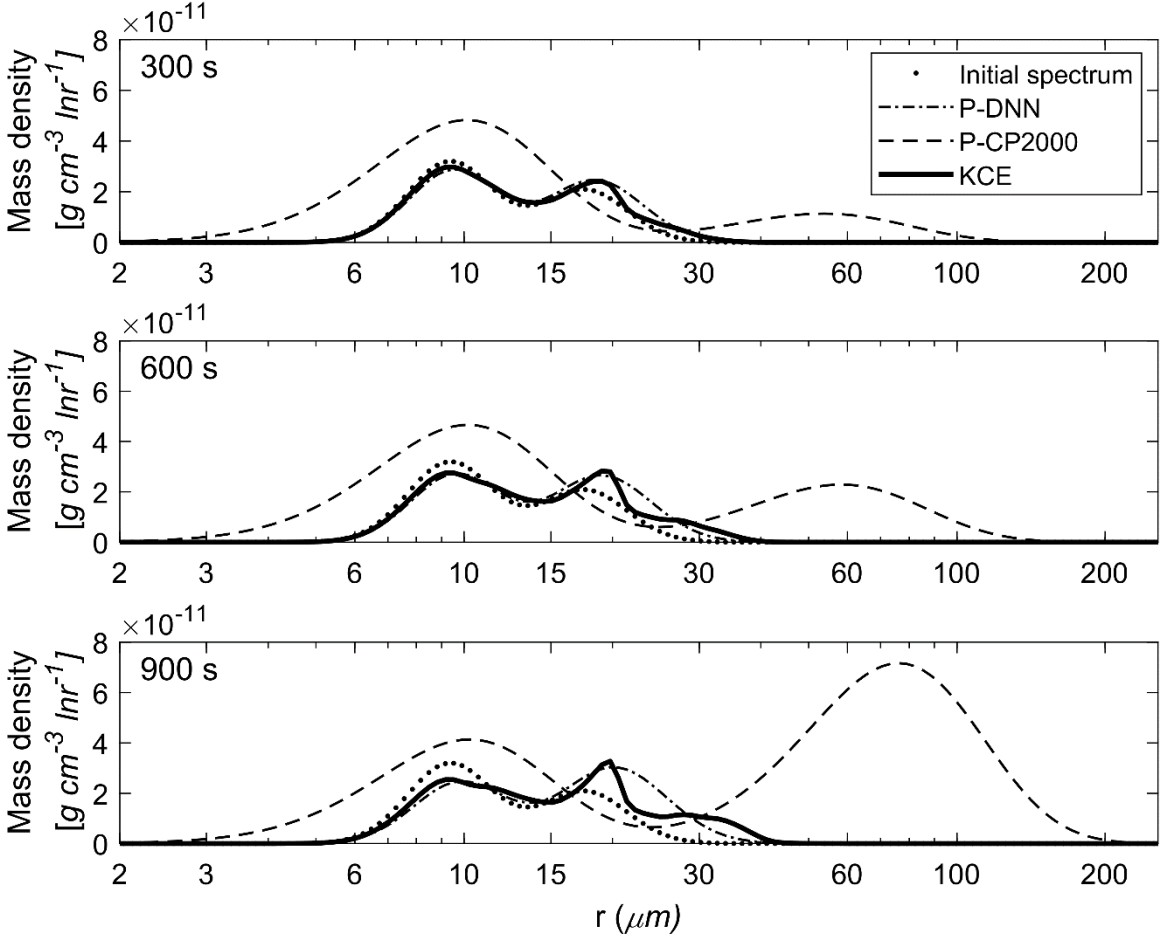

**Figure 7: Mass density functions from P-DNN, P-CP2000 and KCE. The represented times are 300 s, 600 s and 900 s,**
**from top to bottom. Equation (16) was used to transform the drop number concentration spectra from P-DNN to the mass density spectra.**

At 300 s (first row of Fig. 7), there is a slow development of the total spectrum, with a clear mass transfer between both modes of the presented models. The parameter-generated spectrum from P-DNN fits well the reference solution, with a slight overestimation of the maximum mass in the second mode. The mean radius of the distributions are well represented by P-

DNN. At 600 s and 900 s (second and third row of Fig. 7), there is a development of a third mode in the evolution of KCE, that is not reproduced by P-DNN, producing instead a wider second mode, representing well the mean radius and mass

distribution. The first mode is accurately represented at those times. An increase in mean radius can be observed, due to the effect of the collision-coalescence process.

The simulation results with P-CP2000 are clearly different from the others. The first noticeable difference is the existence of droplets that are smaller than the initial distribution. This is caused by the fixed distribution parameters employed in its formulation. The slope parameter is determined by an analytical expression and evolves with time within certain limits, but the parameters related to the spectral breadth are held fixed. For more information please refer to Cohard and Pinty (2000). Besides that, P-CP2000 performs poorly at all the represented simulation times, when compared with KCE. It presents pronounced tendency to go ahead of the KCE, leading to a faster-than-normal development of larger drops. Particularly, the mass transfer is very noticeable at the end of the simulation. However, the first mode of P-CP2000 does not decrease proportionally.

Figure 8 shows a comparison between the drop number concentration spectra derived from P-DNN, P-CP2000 and KCE for three chosen times (300 s, 600 s and 900 s). A generally good agreement is appreciated at all times for P-DNN, with its concentration values slightly underestimating the results from KCE. As the collision-coalescence process decreases the drop number concentration, there is no noticeable increase in the number of drops in the second mode of the distributions. However, an increase in the mean radius is observed, that is consistent with the behaviour described in Fig. 7, where a related mass transfer between both distribution functions is seen.

Regarding P-CP2000, its spectra underestimate the KCE, and the lack of a second mode reaffirms the behaviour shown in Fig. 7. However, being a bulk parameterization, its strong points are not related to the description of the drop spectra, but to the representation of bulk quantities such as the total number concentration and mass content of the clouds.

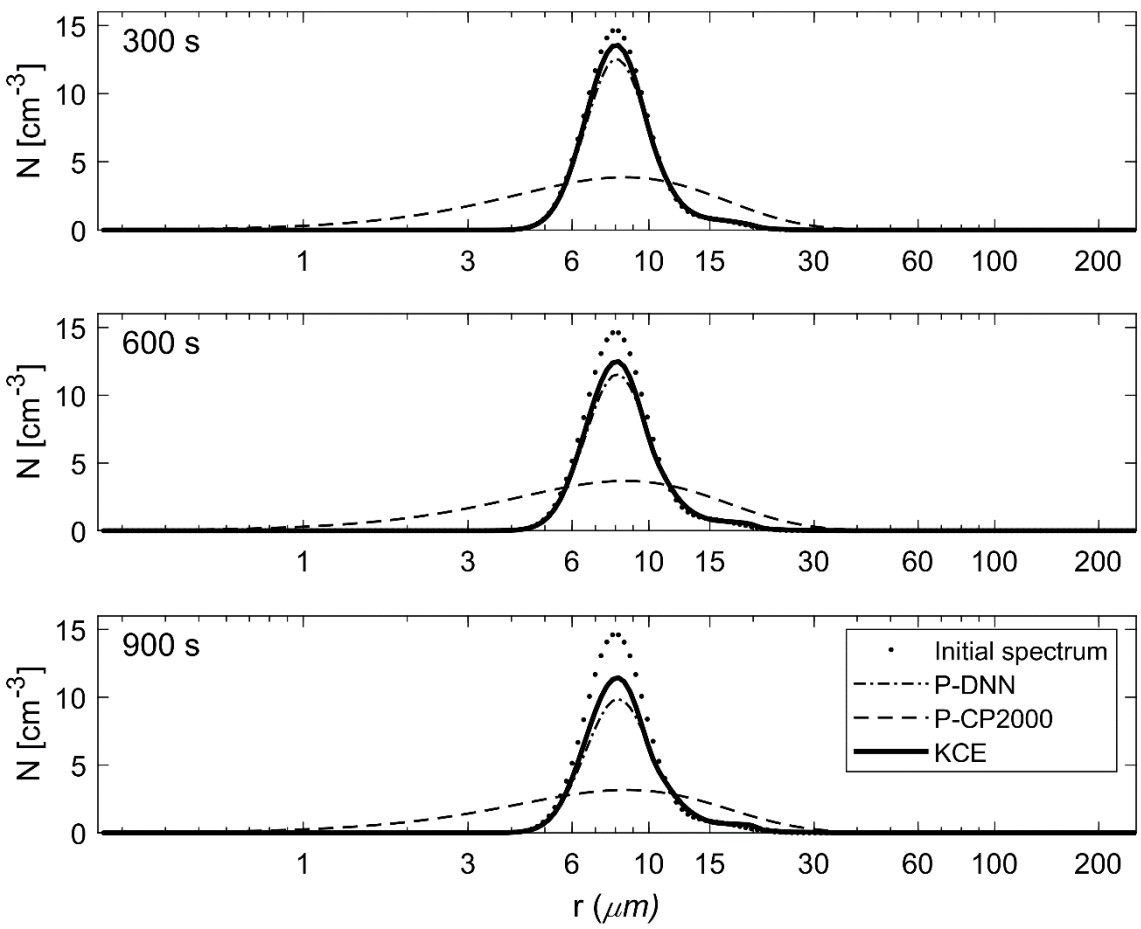

**Figure 8: Drop number concentration spectra for P-DNN, P-CP2000 and KCE. The selected times are 300 s, 600 s and 900 s, from top to bottom. Equation (17) was used to transform the mass density spectra from KCE to the drop number concentration spectra.**

### 5.2 Bulk quantities comparison

Figure 9 shows a comparison of two main bulk quantities (total number concentration and mean radius) obtained from P-DNN, P-CP2000 and KCE. The concentration and mean radius of KCE were obtained by integrating the drop number concentration spectra for the corresponding moment order (0 and 1 respectively). As expected, number concentration decreases with time, due to the coalescence of drops, ranging from an initial value of 200 $cm^{-3}$ to around 160 $cm^{-3}$ in KCE. The predicted concentration from P-DNN underestimates the KCE values throughout most of the simulation time, with the differences reaching 10 $cm^{-3}$ at 900 s. The P-CP2000 model achieves a relatively better representation of drop number concentration, although it reaches the same differences as P-DNN by the end of the simulation.

A similar behaviour is observed in the mean radius results, with a growth in the drop size consistent with the decreasing values on the drop number concentration for P-DNN. However, both P-DNN and P-CP2000 predict the mean radius rather well (considering that their values are diagnosed in both models),  with differences reaching only $0.5\ \mu m$. This result is very important for P-DNN, since radius is the independent variable in the lognormal distributions that act as basis functions for this parameterization. In this regard, P-CP2000 performs somehow worse than P-DNN for the mean radius, with the mean difference almost reaching $1\ \mu m$, although it shows a similar monotony to both the KCE and P-DNN models.

There is a correlation between these results and those depicted in Fig. 8, where concentration decreases with time, and mean radius changes little throughout the simulation, when considering the total moment of order 1. The differences reached in this figure are related to the base model from which the NN model was created. Since the NN model only reproduces (very accurately) the rates of the moments as in Clark (1976), an accuracy improvement of the overall parameterization should not be expected. Despite that, the P-DNN model achieves a physical consistency, and behaves following the rules of the simulated process, as evidenced in the increase on the mean radius (drop growth) and decrease of the number concentration (drop coalescence).

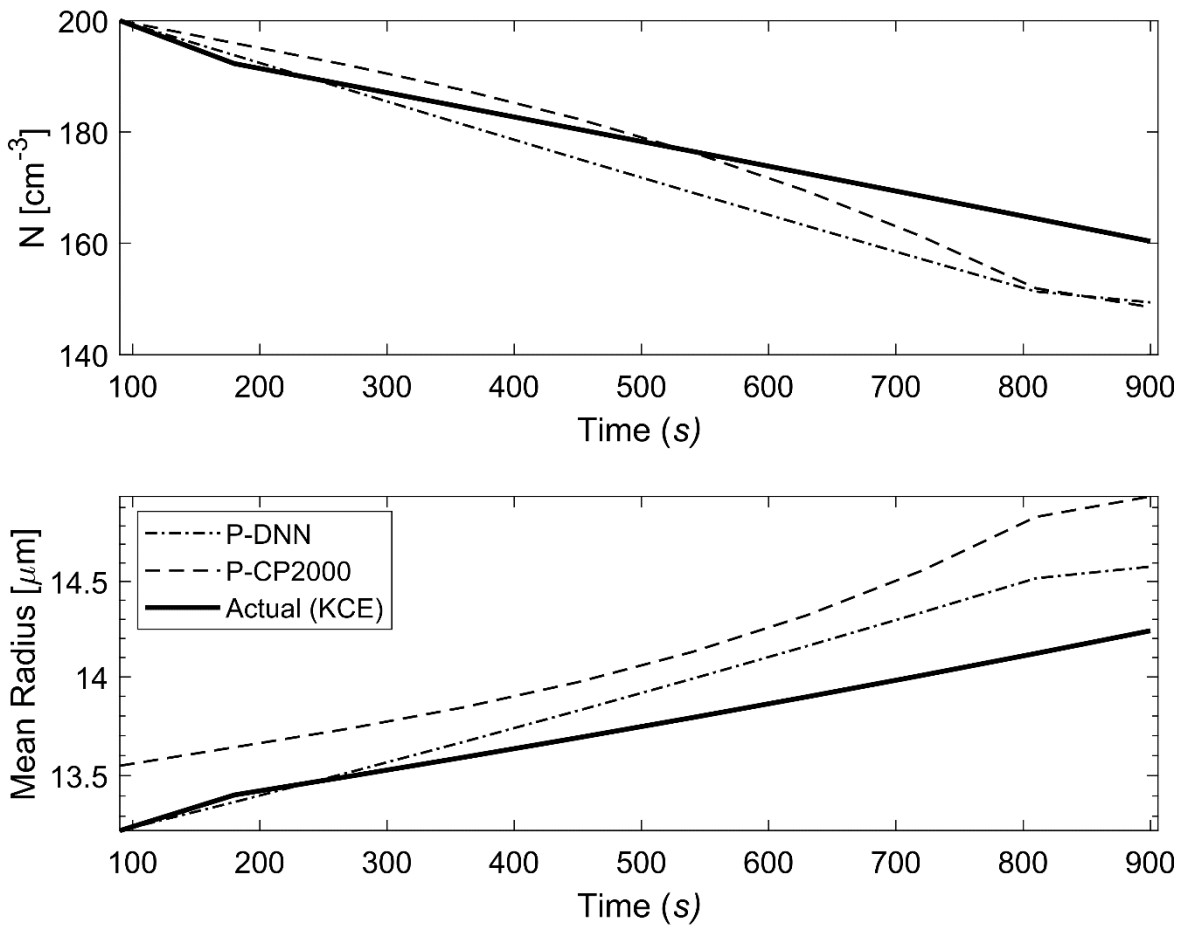

**Figure 9: Drop number concentration (top) and mean radius (bottom) comparison with KCE. The concentration and mean radius of KCE were obtained by integrating the drop number concentration spectra for the corresponding moment order (0 and 1 respectively). The data points are plotted every 60 s.**

Figure 10 depicts the evolution of two main bulk quantities (drop number concentration and liquid water content) for the individual distributions that conform P-DNN ($f_1$ and $f_2$), as well as the combined (total) values of the variables (calculated as $f_1 + f_2$). Regarding concentration, a decrease in $f_1$ values is observed, due to the coalescence process, while a consistent increase in $f_2$ is also appreciated. However, the increase in drop number concentration in $f_2$ is not proportional, because there are a fewer number of bigger drops in the distribution, which are also colliding within the same distribution function (the equivalent of self-collection in bulk parameterizations). This is consistent with Fig. 8, in which the second mode in the concentration DSD is barely developed after 900 s of simulation time. . However, a general decrease of the total concentration value represents well the theory and observations of the parameterized process. Barros et al. (2008) found this same behavior while revisiting the validity of the experimental results obtained by Low and List (1982), excluding drop breakup.

The liquid water content (LWC) values (diagnosed) are depicted only to verify that mass is conserved under the formulation of P-DNN. The LWC of each of the distribution functions ($f_1$ and $f_2$) were obtained from the corresponding moment (order 3) calculated from eq. (4). Effectively, total mass remains constant during the entire simulation, with a proportional mass transfer between $f_1$ and $f_2$. These results demonstrate that the P-DNN parameterization behavior is physically sound, with a remarkable consistency between the different variables calculated, both bulk ($N$, $r$ and $LWC$) and DSD related (concentration and mass density spectra).

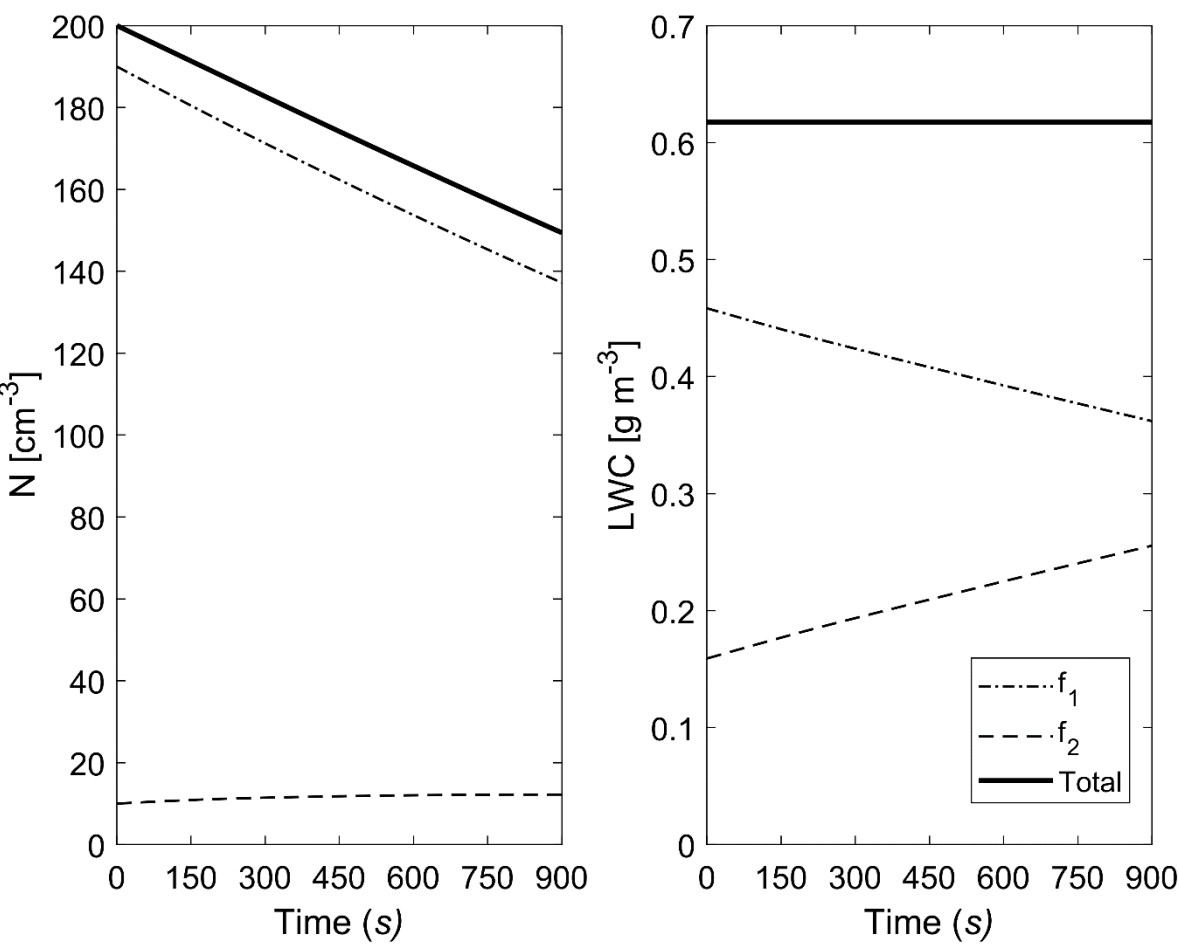

**Figure 10: Evolution of drop number concentration N (left) and liquid water content LWC (right) of the individual distributions that conform P-DNN. The liquid water content of each of the distribution functions ($f_1$ and $f_2$) were obtained from the corresponding moment (order 3) calculated from eq. (4). The combined (total) values of the variables are also shown and were calculated from eq. (5).**

## 5.3 Total moment errors

An analysis of the predicted total moments was performed with the objective to further test the precision of P-DNN, due to the importance of the statistical moments in calculating physical variables such as mean radius and LWC. Table 5 shows the mean percent errors of the total moments predicted by P-DNN and P-CP2000. The percent error is taken relative to the moments of KCE. The data was obtained by calculating the mean of the percent errors of the entire simulation. The moments for the solution of the KCE were computed by integrating the reference drop number concentration spectra using eq. (3), while the total moments from P-DNN and P-CP2000 were calculated using the predicted distribution parameters and solving eq. (5). The defined gamma distribution equations for the moments are used in the case of P-CP2000. A reasonable degree of accuracy was achieved by P-DNN, with the mean error never surpassing the 4 %. However, the data shows that the total moments of order 0 to 2 are usually underestimated, while those of order 4 and 5 are slightly overestimated. This could result in the calculations of drop number concentration values lower than the actual ones, as seen in Fig. 9. As for P-CP2000, the model is not formulated to predict individual moments other than the zeroth and third moments. Thus, the other moments of the distributions are not well represented, as observed in the mean percent error, which reaches almost -61% for the moment of order five. That value is a great difference from the zeroth moment for example, whose percent error is only -1.3 %. This result indicates that P-DNN is adequate to represent the evolution of individual moments within certain ranges, when compared with more conventional bulk schemes.

**Table 5: Total moment mean errors. The percent error is taken relative to the moments of KCE. The shown data was obtained by calculating the mean of the percent errors of the entire simulation.**

| Total Moment Order | Mean Percent Error P-DNN | Mean Percent Error P-CP2000 |
| --- | --- | --- |
| M0 | -3.3479 | -1.2718 |
| M1 | -2.6437 | 27.0500 |
| M2 | -1.4969 | 27.0370 |
| M4 | 1.1249 | 9.2037 |
| M5 | 0.7205 | -60.8886 |

Figure 11 shows the time evolution of the percent error of the total moments throughout the simulation for P-DNN. The percent error is taken relative to the moments of KCE. The moments of KCE were calculated by integrating the reference drop number concentration spectra using eq. (3), while the total moments from P-DNN were calculated using the predicted distribution parameters to solve eq. (5). The error of total moment of order 3 is zero during the entire simulation because mass is not affected by the collision-coalescence process. The total moments from order 0, 1 and 2 overestimate the KCE in the first 300 s of simulation, underestimating them for the rest of the P-DNN run, with the percent error reaching a minimum value of -8 %. The opposite behaviour is appreciated for the total moments of order 4 and 5, where they initially underestimate the KCE, overestimating it for the rest of the simulation. However, for these orders the percent error is usually lower, with a maximum

of 4 %. Generally, P-DNN is performing well, with the percent error never reaching the 10 % threshold. However, further analysis on this topic is recommended, to improve the accuracy of the parameterization.

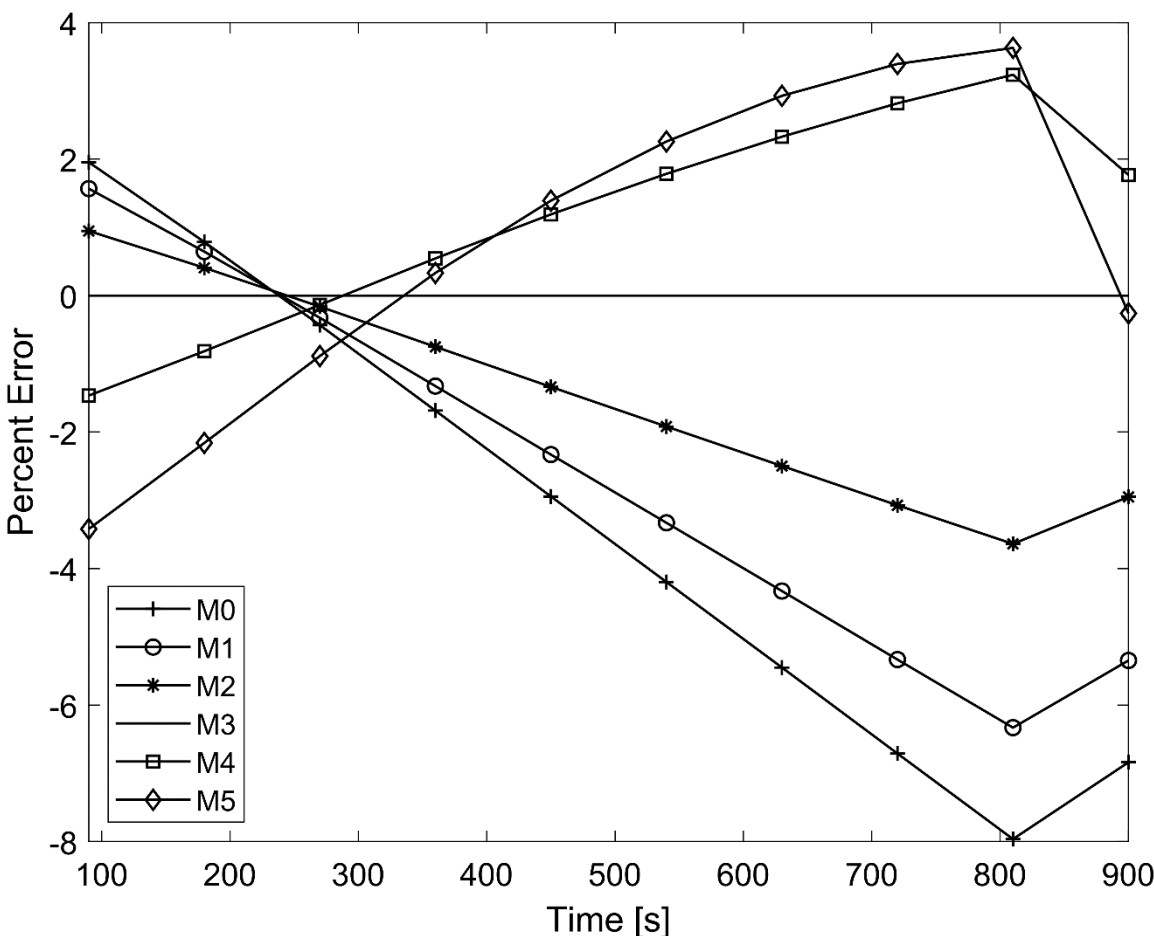

**Figure 11: Time evolution of the errors corresponding to the predicted moments from P-DNN. The percent error is taken relative to the moments of KCE. The moments of KCE were calculated by integrating the reference drop number**
**concentration spectra using eq. (3), while the total moments from the P-DNN were calculated using the predicted distribution parameters to solve eq. (5).**

## 6 Conclusions

The presented way to simulate the evolution of the droplet spectra due to collision-coalescence falls within the framework developed by Clark (1976) and Clark and Hall (1983). Under this approach, a dynamic framework has been established in
Rodríguez-Genó and Alfonso (2021e). A hybrid parameterization for the process of collision-coalescence based on the methodology of basis functions employing a linear combination of two lognormal distributions was formulated and

implemented. All the parameters of the distributions are derived from the total moment tendencies, which in turn are calculated by means of five trained deep neural networks. By doing this, we obtained a parameterized model that determines the distribution parameters' evolution, hence, the evolution of the DSD. The physical variables are diagnosed from the distribution moments. Within the framework of this parameterized model, there is no artificial classification of the water substance (cloud droplets or raindrops). Instead, we consider a full set of distribution parameters for each of the distribution functions considered in the formulation of the parameterization, in order to describe the DSD in radius space. This kind of microphysical parameterization allows the use of an arbitrary number of probability density functions in linear combination to reproduce the drop spectrum.

The novel components of the P-DNN model were introduced and evaluated in Chapter 3. A total of five Deep Neural Networks were trained to calculate the rates of the total moments following Clark (1976), using a novel training approach called cascade-forward neural network, instead of the traditional sequential networks. By doing this, the trained NNs were able to accurately reproduce the total moment tendencies, showing a very high correlation and small MSE , when compared with those calculated with the original formulation (Clark, 1976) (See Figs 5 and 6, and Table 3). Thus, it was demonstrated the precision and ability of the ML-based method to reproduce the rates of the total moments due to collision-coalescence, when trained with a sufficient number of data samples (100000 combinations).

One experiment was performed to illustrate the behaviour of the DNN-based parameterization at the initial stages of cloud formation. The simulation results from P-DNN showed good agreement when compared to a reference solution (KCE), for both the predicted DSD and the bulk quantities considered. Moreover, the P-DNN model demonstrated a physically sound behaviour, adhering to the theory of the collision-coalescence process, with overall consistency in the values of both prognosed and diagnosed variables. With the development of P-DNN, a parameterization for solving the entire collision-coalescence process have been developed (with exception of drop breakup) using a ML methodology. To the best of the authors' knowledge, previous attempts on describing collision-coalescence using the same methodology have been focused on super-parameterizations for sub-grid processes (Brenowitz and Bretherton, 2018), or formulations for specific sub-processes such as autoconversion (Alfonso and Zamora, 2021; Loft et al., 2018) and accretion (Seifert and Rasp, 2020).

For comparison purposes, the bulk parameterization developed by Cohard and Pinty (2000) (P-CP2000) was also implemented. According to the comparison with the bulk model, the main strength of P-DNN is the superior ability to represent the evolution of the total moments and the shape of the DSD, because of its formulation based on time-varying distribution parameters. The predicted P-DNN concentration and mass spectra closely match that of the reference bin model (KCE), while showing a good accuracy at forecasting bulk variables such as drop number concentration and the parameter-derived mean radius. Furthermore, total mass is conserved throughout the entire simulation, which is remarkable due to the inherent numerical diffusion of the first order finite differences method applied to update the parameters at each time step. An analysis of the accuracy of the predicted total moments of P-DNN was performed, with the percent error relative to the KCE never exceeding 8 %. However, there is room for improvement in the calculations of the total moments. Thus, it is the recommendation of the authors to retrain the DNNs with a finer resolution in the parameters' values, and with a wider range of values in order to cover all possible

combination of parameters. Another recommendation is the inclusion of the drop breakup process in the formulation, which is currently not included in the parameterization, despite its influence on the behaviour at the edge of raindrop distributions. In addition, the use of ML eliminated the requirement of numerically integrating the total moment tendencies at each time step, and the use of lookup tables for each predicted moment is no longer needed under this formulation.

To obtain a full warm cloud model, an extension of this neural network algorithm applied to condensation is proposed, following the same methodology of series of basis functions. A parameterization scheme such as this could be included in regional weather and climate models, as its initial conditions can be calculated from variables needed by more traditional bulk models.

### Author contributions

Léster Alfonso performed the conceptualization of the article, designed the methodology, acquired funding and resources, supervised, reviewed and edited the original draft. Camilo Fernando Rodríguez Genó realized the formal analysis, investigation, developed and implemented the software and code for simulation and visualization, validated the results, and wrote the original draft preparation.

### Code availability

The current version of *COLNET (COLNETv1.0.0)* used to produce the results presented in this paper is archived on Zenodo (https://doi.org/10.5281/zenodo.4740061) (Rodríguez-Genó and Alfonso, 2021a) under the *GNU Affero General Public License v3 or later* licence, as well as all needed scripts to run the model. The outputs of the model used to generate the figures included in the present paper are also included. The scripts used in the generation of training data sets and for training the neural networks used in *COLNETv1.0.0* (Rodríguez-Genó and Alfonso, 2021c) can be found on Zenodo
(https://doi.org/10.5281/zenodo.4740129), while the codes for plotting the figures (Rodríguez-Genó and Alfonso, 2021b) are stored at https://doi.org/10.5281/zenodo.4740184. The original FORTRAN77 code of the explicit bin model is archived at https://doi.org/10.5281/zenodo.5196706, and have been licensed (*GNU Affero General Public License v3 or later*) and versioned (V1.0.0) with the permission of the author. The code used for the WDM6 parameterization simulation (Rodríguez-Genó and Alfonso, 2021d) can be found on Zenodo (https://doi.org/10.5281/zenodo.5196706). The models and related scripts
were written using MATLAB R2020a, with exception of the explicit model.

### Competing interests

The authors declare that they have no conflict of interests.

## Acknowledgments

Camilo Fernando Rodríguez-Genó is a doctoral student from Programa de Posgrado en Ciencias de la Tierra at Universidad Nacional Autónoma de México, and received fellowship 587822 from Consejo Nacional de Ciencia y Tecnología (CONACyT). This study was funded by grant no. CB-284482 from the Consejo Nacional de Ciencia y Tecnología (SEP-CONACyT).

## Financial support

This research has been supported by the Consejo Nacional de Ciencia y Tecnología (CONACyT) by means of grant no. CB-284482 and fellowship no. 587822.

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
