# Peer review of "Parameterization of the collision-coalescence process using series of basis functions: *COLNETv1.0.0* model development using a machine learning approach"

_Geoscientific Model Development, 2021_

## Referee Comment (RC2)

**Review:**
**Parameterization of the collision-coalescence process using series of basis functions: COLNETv1.0.0 model development using a machine learning approach**

Camilo Fernando Rodríguez-Genó, Léster Alfonso

This study introduces a parameterization of the collision-coalescence process of cloud droplets, which models the cloud droplet size distribution as a sum of two lognormal distributions. Within this parameterization, the time rates of change (tendencies) of five moments of the distribution are predicted by a deep neural network – traditionally, these moments are either computed by solving an integral or by using pre-computed lookup tables.

The new parameterization is compared against a reference solution, the explicit model developed by Bott (1998). In this comparison, the distribution moments obtained from the new parameterization deviated by less than 10% from those of the reference solution.

**General comments**

I think that the core idea of the paper, to replace a computation in the simulation of the collision-coalescence process with the predictions of a machine learning model, is a valid one. However, I have a few major concerns and questions:

- First of all, the manuscript is in need of some thorough editing for clarity and correctness. There are plenty of grammar mistakes, typos, and confusing phrasing, such that it is overall not pleasant to read.
- Given that the machine learning application presented here is very straightforward (the training data cover all possible parameter ranges the model will encounter in the experiment, so all the model has to do is to learn how to interpolate the training data, there is no generalization needed beyond what it has already seen), I would have wanted to see a better justification of its utility. More concretely: How much time and/or memory is saved by the DNN compared to directly

computing the moment tendencies (Eq. 13) using a numerical integration method such as a trapezoidal rule, and compared to using a lookup table for these integrals (the introduction mentions that this is a commonly used method)? It would also be interesting to see how these time savings compare to the total runtime of a typical simulation (since runtime optimization should aim at the computational bottlenecks).

For example, a lookup table of the size of the dataset used here can fit in a Level 3 cache (if I understand correctly, 1'000'000 samples, so 1'000'000 x 5 targets were generated in total – assuming each target is a 64 bit (8 byte) float, we get a total size of about 40 MB), so it might well be that the lookup table is faster than the DNN predictions (but of course, it requires more memory, and it only contains moment tendencies for a pre-defined set of input values whereas the DNN will predict on any given input). Without estimates of the trade-offs (accuracy, speed, memory demand) involved, it is impossible to see the added value of using a machine learning model for the task of predicting the moment tendencies.

- I think the study would be stronger if the new parameterization was not just evaluated for a single experiment, but for several experiments with different initial conditions, maybe even exploring some of the "edge cases" (e.g., what happens when the number of drops approaches 1, which is the state any collision-coalescence process will converge to?)

**Specific comments**

- L 9: "drop spectrum", not "drop spectra" (it's singular)
- L 15: "stablish" should probably be "establish"
- L 23: "who used" instead of "whom employed"
- L 24: "has shown", not "have shown"
- L 27: "For spherical particles such as cloud drops, a transformation of the DSD leads to a self-preserving form" – can you briefly explain what this means? Also, it is unclear how this and the following two sentences connect to the previous sentence, which highlights the superiority of the lognormal distribution in terms of squared-error fit compared to gamma or exponential distributions.

- L 28: Maybe remind the reader of the definition of the Knudsen number and its implications for the validity of the continuum assumption of fluid mechanics?

- L 25 – 34: I find the purpose of this whole segment unclear and its phrasing confusing. Is the idea to underline the suitability of the lognormal distribution to the modeling of cloud droplet size distributions? If so, please make this more explicit and state when a sentence is specifically about lognormal distributions. E.g.,"The analysis of […] showed that the lognormal distribution adequately represents the particle distributions" seems to be aimed at strengthening the case for the lognormal distribution as an adequate description of DSDs (it needs a citation though), whereas the following sentence ("Further, …") seems to be a general statement about the dependence of the rate of convergence on the initial geometric standard deviation.

- L 36: The abbreviation DSD has already been introduced in L 21.

- L 44: "need to calculate a huge amount of equations, which number ranges from several dozens to hundreds, at each grid point and time step" –> "need to calculate dozens to hundreds of equations at each grid point and time step"

- L 44: Also mention numerical diffusion as one of the major problems with bin microphysics? See e.g. [1]

- L 57: "20 µm and 41 µm being" instead of "being 20 µm and 41 µm" – **I won't continue to do "micro-corrections" of grammar and typos, but the manuscript really needs some thorough editing for clarity and correctness (see my first general comment). Not being a native English speaker myself, I do understand the difficulty of writing in a foreign language, but putting some effort into this will result in a more reader-friendly paper that stands a better chance of getting read and cited by other scientists.**

- L 91: This introduction to machine learning seems kind of out of place, especially after the previous paragraph already talks about deep neural networks.

- General remark about equations: Please define all variables involved, even if their meaning seems straightforward – e.g., in Eq. (1), say that r is radius, in Eq. (2), say what N is, etc.

- Neural network architecture: How did you come up with this specific architecture? Did you try other (e.g., simpler) architectures as well?

- L 171: The commonly used terminology in machine learning is that the training data are the data used to fit the model, the validation data are used for model selection (e.g., when you are testing different neural network architectures, or comparing, say, the neural network with a

random forest model, you decide on a final model based on the models' performances on the validation data), and the test set is used for assessment of the generalization error of the final chosen model (see e.g. [2]). Since no model selection is done in this study, what is the called "validation set" should more appropriately be called the test set here.

- L 214: How were the ranges of the µ and σ parameters (rightmost column of Table 1) for the uniformly random sampling of the distribution parameters that was used to generate the training data determined? Were they "reverse engineered" based on a certain range of LWC values that are thought to be physically reasonable?

- L 234: I think it would be interesting to include the collision-coalescence parameterization using the trapezoidal rule to solve Eq. (13) in the results (e.g., in Figure 8) – presumably the main advantage of predicting the moment tendencies using the DNN rather than computing them using the trapezoidal rule is computational efficiency, so it would be nice to know how much faster the DNN is, as well as to see how the mass density spectra obtained using this "trapezoidal parameterized model" compare to those shown in Figure 8 (reference solution and predicted parameterized model). See also my second general comment. Based on the good agreement between the DNN predictions and the validation targets computed using the trapezoidal rule (Figure 7), the resulting mass density spectra will probably look very similar, but I think it would still be interesting for the reader to see that comparison.

- L 336: I think it's a bit of a stretch to say that the third mode in the evolution of the KCE-generated spectra "is reproduced by the parameterization as a wider second mode" – it seems to me that the parameterization is not able to capture that development.

**Figures**

- 7: The x axis label ("Actual Total Moment Tendencies") of M0 and M1 are missing

**References:**

[1] Khain et al., 2000 , A., M. Ovtchinnikov, M. Pinsky, A.Pokrovsky, and H. Krugliak (2000): Notes on the state-of-the-art numerical modeling of cloud microphysics. Atmos. Res., 55,159–224, https://doi.org/10.1016/S0169-8095(00)00064-8.

[2] Hastie, T., Tibshirani, R.,, Friedman, J. (2001). The Elements of Statistical Learning. New York, NY, USA: Springer New York Inc..

---

## Author Response (AR1)

The authors thank the anonymous referees for their helpful comments that improved the quality of the manuscript.

**Comments from Anonymous Referee # 1 and answers from the authors**

*General comments*
*The study introduces a new parameterization of the collision-coalescence process that is based on the results from machine-learning procedures, with an aim to eventually use it in weather forecasting models. The authors utilized 100,000 size distributions of drops (including both cloud droplets and raindrops) to obtain the tendencies (time derivative) of 0th-5th moments, which were used for training a machine (80%) or evaluating the machine's predictions (20%). Each droplet size distribution was assumed to be a composite of two lognormal size distributions, represented by 6 parameters. The paper compares the evolutions of drop size distributions predicted by the machine-learning based parameterization and explicitly calculated by the method in Bott et al. (1998). The authors concluded that the differences were always less than 10% and therefore it has a promising potential for the future implementation in weather forecasting models.*
*The overall idea of utilizing the machine-learning method is innovative and aligns with what the cloud-modeling community has started working on in recent years. The results of the study are interesting and provide promising suggestions for the future model improvements. At the same time, the paper seems to require some improvements in its structures and also in providing sufficient information. Most importantly, the conclusions would become much more solid and significant if (i) more than one test simulation is done and/or (ii) if the comparison to an existing parameterization is shown. Regarding (i): although a large number of samples were used for training the machine, the overall evaluation of the new parameterization seems to rely only on one simulation (Table 4), particularly its comparison with the explicit calculation by Bott et al. (1998) under the same condition. The prediction accuracy must be somewhat dependent on each case and it is not known if this one test case falls in the "well-" or "badly-" predicted group. Regarding (ii): the prediction would always have some errors, but the magnitude of the errors is important, particularly in comparison to errors made by other existing parameterizations. Therefore, I think (i) more test simulations to compare the predictions with Bott's calculations and/or (ii) comparison with existing two-moment parameterization is necessary to draw a solid conclusion. I would highly suggest (ii). Detailed suggestions are listed below.*

Answer: Regarding (i), the authors agree with the referee on performing more test simulations. However, it is not the objective of the paper to show the behavior of the parametrization under several initial conditions, or even under extreme cases of study, but to introduce the Machine Learning methodology applied to the series of basis functions modelling philosophy, and to eliminate the need to solve complex integrals as part of the formulation of the parameterization. Further testing will be done addressing those and more concerns, including the addition of a condensation module.

Regarding (ii), an additional comparison has been included in the revised version of the manuscript, taking into account an extra parameterization, as suggested by the referee. The

popular WDM6 (WRF Double Moment 6-class) parameterization was used in the simulation, using the same initial conditions and simulation parameters. The results and discussion of the comparison have been included in the updated version of the manuscript. It was the intention of the authors to include a second extra parameterization in the paper ((Seifert & Beheng, 2001)), but because deadline issues and the extensive work needed, it was not included.

***Specific comments***

*Lines 12-13: It seems very important to clarify what was calculated and what was predicted/estimated. Since it's supervised learning, the machine did not calculate the moments based on equations, but they must have been calculated in advance elsewhere and the results (inputs & output) were fed into the machine to train it. Afterwards, during the testing/validation phase, the total moments were predicted, not calculated by physical equations, by the trained machine. I understand the overall meaning but the readers may be misled that the machine can analytically solve the SCE and calculate the tendencies of the moments. But in reality, the machine simply gives the prediction based on what it learned before. Therefore, the word "predict/estimate" sounds more appropriate than "calculate".*

Answer: The authors agree with the referee, and the wording of the abstract have been changed to reflect the fact that the Machine Learning model only predict the tendencies of the total moments, and does not solve the SCE itself.

*Line 27: Adding a short explanation on a self-preserving form would be helpful (e.g., what it is, why it gets formed, etc.), especially if this is relevant to collision-coalescence.*

Answer: The self-preserving form size distributions are analyzed in detail on (Swift & Friedlander, 1964), and is related to the preservation of the type of distribution function with time. Self-preserving distributions are relevant to collision-coalescence mainly because the evolution of the distribution functions due to this process can be expressed in this mathematical form.

*Section 2: The structure of this section would become better if it's modified, so that there are 2.1 and 2.2, instead of only 2.1. In my observation, the first section in 2 (that I suggest to convert to 2.1) is dedicated to the time derivative of moments, regardless of collision-coalescence. Subsection 2.1 (that I suggest to change to 2.2) is providing the SCE. Mathematically speaking, I had hard time connecting the two, Eqs. 6 and 13, as Eq. 6 is not mentioned later in the paper, although I understood/knew them individually. Therefore, I suggest that the authors add a few sentences at the end of Section 2 to summarize the entire section.*

Answer: The structure of the section have been modified to better organize the contents, rearranging the subsections as 2.1 and 2.2. The system of equations expressed in Equation 6 is transformed to its matrix form in Eq. 7. Equation 13 represents the way on which the total moment tendencies are calculated in the original parameterization (Clark, 1976), and is the definition of the components of vector **F** (right-hand side of the system of equations).

*Lines 211-212: Although mentioned later, it would be better to mention here why the third moment tendency is not calculated.*

Answer: An explanation is made about why the third moment order is not included, as suggested by the referee.

*Figure 4: The figure would be more helpful if the authors instead provide a distribution (line or bar plots) of all the data rather than a scatter plot of every 100 data. Moreover, if the information (e.g., minimum, maximum, mean, median, etc.) can be provided separately for two lognormal distributions on Table 1, this figure can be omitted, as the information overlaps.*

Answer: The authors agree with the redundancy of information between Figure 4 and Table 1. Thus, Figure 4 has been deleted from the article, and the rest of the figures have been renumbered.

*Table 3: If the authors can add a column for a prediction score, that would be helpful too, if Matlab has a function to calculate prediction scores. The actual values of MSE may be difficult for the readers to assess the accuracy of the prediction. For example, in the text, MSEs on the other of 10-4 are considered to be a good performance, but could you explain this assessment in more detail? For example, above what number is considered a poor performance, and why, etc.*

Answer: Since the values of the total moment tendencies are normalized (scale of $10^0$), MSE values of $10^{-4}$ are considered a good performance. This explanation has been included in the manuscript, for more clarity in the text and interpretation of results. A column has also been included in Table 3, detailing the Correlation Indexes calculated between the output of the trained neural networks and the solution of the KCE.

*Section 4: I think this section can be included as a subsection of 5.1 in the following Section 5, or even as 2.3 in Section 2.*

Answer: The authors agree with the suggestion of the referee, and Section 4 has been relocated as subsection 2.3. All subsequent equations and sections have been renumbered accordingly.

*Table 4: I understand that these conditions were chosen based on Clark (1976), but I think it would strengthen the argument that this case (or f1) is a good representation of the training data on which the machine was trained, if the authors mention the mean values in Table 1.*

Answer: An explanation was included in the manuscript to reflect the fact that the initial conditions from Table 4 are in fact a good representation of the data used to train the neural networks.

*Lines 357-358 and Figures 9 and 10: It is difficult to conclude whether the differences between what's predicted by the new parameterization and what's calculated by Bott's code are small enough or not, only from the figures. However, if you can add predicted values from other existing two-moment parameterizations (one frequently used in weather forecasting models), that would give the readers some insight; in Figure 10, for example, if another parameterization predicts 100 cm-3 at t=900s, then the new machine learning-based parameterization would be a better predicter. Furthermore, if such a comparison can be done for more than one case, the results would become much more solid and substantial.*

Answer: In order to better demonstrate the accuracy of the developed parameterization, a comparison with the results from the collision-coalescence section of the WRF Double Moment 6-class parameterization (WDM6) have been established (Cohard & Pinty, 2000). However, a comparison methodology had to be developed, since both parameterizations are of different kinds, and their formulations are focused on different modelling philosophies. Despite that, the comparison showed promising results for the Machine Learning parameterization, particularly in the calculation of the individual moments of the drop spectrum. The proper figures and comments have been added to the manuscript, to incorporate those new findings from the comparison. It was the intention of the authors to compare the results with at least another parameterization (the one from (Seifert & Beheng, 2001)), but the amount of work needed to establish that comparison exceeded the available time offered by GMD, due to the extensive differences between the formulations of the parameterizations. Such work will be done in future research regarding the parameterization philosophy of  series of basis functions here presented.

*Table 5 and Figure 12: While the authors clearly state the percentage differences between the predictions and the explicit calculations, its physical meaning also needs a clarification. For example, what does the -8% error of M2 tendency prediction physically mean, and why could it be underestimated by the machine? Even more, for instance, how does this magnitude of errors compare to the errors made by other existing parameterizations?*

Answer: The calculation of the percent errors are done taking the bin model results as reference. For example, a -8 % error of M2 tendency means that the predicted value of that specific moment is 8 % lower than the reference solution, regarding the reference solution itself. The causes of those differences are still subject of investigation. However, the comparison with one commonly used parameterization (explained in the previous answer) shows a better skill at predicting the statistical moments of the drop spectra than the added parameterization (WDM6). To reflect this, Table 5 has been modified to include the results of the extra parameterization considered.

*Section 7: The authors conclude that the overall prediction accuracy was high, but additional analyses and/or a comparison with existing parameterizations seems to be necessary to draw the conclusion. Although the errors in Figure 12 remained less than 10%, how about other existing parameterizations? Would they be within 5%, or more than 50%? I think such a comparison would provide the readers more in-depth understanding and better assessments of the presented ML-based parameterization.*

Answer: Same as the two previous comments. The authors understood that comparison with at least one extra parameterization was needed in order to provide a better assessment on the accuracy of the Machine Learning model.

**Technical corrections**

The authors thank the referee for the detailed revision of the technical details of the manuscript. All recommendations have been addressed, and we will only answer the ones that required specific comments.

*Lines 70-72: As it approximates the droplet size distributions by two lognormal distributions, rather than using bins, I am not sure if "This approach simulates the explicit approach" is the accurate description. The strength of the authors' approach seems to be the time-varying parameters for the two lognormal distributions, in contrast to the conventional bulk schemes, which can be emphasized here.*

Answer: As noted by the referee, the strength of the presented parameterization resides in the time-varying parameters for the distributions. However, is the authors' opinion that this approach could be considered a middle point between bin and bulk models, as it covers the entire size spectrum with continuous, non-truncated, distribution functions. However, we have followed the recommendations of the referee of emphasizing the main characteristic of the parameterization.

*Figure 7: Since the values from the explicit calculations are the "goal/right" values, I think they should be plotted on the y axis rather than on x (i.e., suggest swapping x and y axes).*

*Also, the plots would look better if the x- and y ranges are identical within each plot (e.g., the plots for M1 and M4 seem to have different ranges for x and y axes).*

Answer: The values from the Neural Network model are plotted in the y axis to achieve consistency across all figures in the manuscript. Since all results from the parameterization are plotted in the y axis, the authors consider that Figure 7 (renumbered Figure 6 in the revised manuscript) should not be the exception.

Regarding the ranges of the axles, while it is true that the plots would look better if the axles were identical, it is necessary to reflect that each moment has different ranges according to their characteristics. Since the values of the moments' rates are not normalized, the axles cannot be in the identical for all plots in Figure 7.

*Figure 11: Though this is a small point, it would be better for the two panel plots to be placed top-and-bottom instead of left-and-right, as they share the x axis.*
Answer: Following the same logic of the referee, it was the first intention of the authors to place the figure in the indicated way, prior to submission to the journal. However, after reviewing the manuscript, we noted that that configuration caused the plots to be deformed and the results could not be easily interpreted, so we opted for a left-and-right configuration of the panels.

**Comments from Anonymous Referee # 2 and answers from the authors**

*General comments*

*I think that the core idea of the paper, to replace a computation in the simulation of the collision coalescence process with the predictions of a machine learning model, is a valid one. However, I have a few major concerns and questions:*
*• First of all, the manuscript is in need of some thorough editing for clarity and correctness. There*
*are plenty of grammar mistakes, typos, and confusing phrasing, such that it is overall not pleasant to read.*
*• Given that the machine learning application presented here is very straightforward (the training*
*data cover all possible parameter ranges the model will encounter in the experiment, so all the*
*model has to do is to learn how to interpolate the training data, there is no generalization needed*
*beyond what it has already seen), I would have wanted to see a better justification of its utility.*
*More concretely: How much time and/or memory is saved by the DNN compared to directly*
*computing the moment tendencies (Eq. 13) using a numerical integration method such as a trapezoidal rule, and compared to using a lookup table for these integrals (the introduction mentions that this is a commonly used method)? It would also be interesting to see how these*

*time savings compare to the total runtime of a typical simulation (since runtime optimization should aim at the computational bottlenecks).*
*For example, a lookup table of the size of the dataset used here can fit in a Level 3 cache (if I*
*understand correctly, 1'000'000 samples, so 1'000'000 x 5 targets were generated in total – assuming each target is a 64 bit (8 byte) float, we get a total size of about 40 MB), so it might well be that the lookup table is faster than the DNN predictions (but of course, it requires more*
*memory, and it only contains moment tendencies for a pre-defined set of input values whereas the DNN will predict on any given input). Without estimates of the trade-offs (accuracy, speed,*
*memory demand) involved, it is impossible to see the added value of using a machine learning model for the task of predicting the moment tendencies.*
*• I think the study would be stronger if the new parameterization was not just evaluated for a*
*single experiment, but for several experiments with different initial conditions, maybe even exploring some of the "edge cases" (e.g., what happens when the number of drops approaches*
*1, which is the state any collision-coalescence process will converge to?)*

Answer:

- Regarding the need of thorough editing, we have performed a major review of the grammar and phrasing, thanks to the helpful comments of both referees, and the reviewed version of the manuscript should have improved in quality.
- Regarding the justification of the utility of the machine learning parameterization, it relates to the more straightforward way of computing the moment tendencies mentioned in the manuscript. Since the numerical solution of eq. (13) is a complex task, particularly the selection and implementation of an efficient numerical method or quadrature for the solution of double integrals, and as the use of lookup tables is a popular but less-than-ideal solution of the problem, the objective of the manuscript is to find an alternate way of computing the rates of the total moments, without sacrificing precision. An exhaustive computational or hardware-focused analysis of the problem falls outside the scope of the presented paper, since the performance of the parametrization depends specifically of the computational platform employed to run the simulation, and the characteristics of the hardware. Besides, as the model is not coded in parallel, it would make no sense to evaluate those characteristics, because it would not be using the full potential of the computational platform employed, and the distribution of the processors (including caches) and memory flow is in a single way.
- Regarding the realization of new experiments, the authors agree with the referee on performing more test simulations. However, it is not the objective of the paper to show the behavior of the parametrization under several initial conditions, or even under extreme (edge) cases of study, but to introduce the Machine Learning methodology applied to the series of basis functions modelling philosophy, and to

eliminate the need to solve complex integrals as part of the formulation of the parameterization. Further testing will be done addressing those and more concerns, including the addition of a condensation module to the parameterization.

***Specific comments***

- *L 9: "drop spectrum", not "drop spectra" (it's singular)*

  Answer: The error has been fixed.

- *15: "stablish" should probably be "establish"*

  Answer: The error has been fixed.

• L 23: "who used" instead of "whom employed"

Fixed

• L 24: "has shown", not "have shown"

Answer: Fixed

• L 27: "For spherical particles such as cloud drops, a transformation of the DSD leads to a self preserving form" – can you briefly explain what this means? Also, it is unclear how this and the

following two sentences connect to the previous sentence, which highlights the superiority of

the lognormal distribution in terms of squared-error fit compared to gamma or exponential

distributions.

Answer: The order of the sentences in that first paragraph of the Introduction was mixed. The entire paragraph has been restructured and now makes more sense for the reader.

• L 28: Maybe remind the reader of the definition of the Knudsen number and its implications for

the validity of the continuum assumption of fluid mechanics?

Answer: A brief explanation of the Knudsen number and its implications on the problem at hand has been added to the introduction.

• L 25 – 34: I find the purpose of this whole segment unclear and its phrasing confusing. Is the

idea to underline the suitability of the lognormal distribution to the modeling of cloud droplet

size distributions? If so, please make this more explicit and state when a sentence is specifically

about lognormal distributions. E.g.,"The analysis of […] showed that the lognormal distribution

adequately represents the particle distributions" seems to be aimed at strengthening the case for

the lognormal distribution as an adequate description of DSDs (it needs a citation though),

whereas the following sentence ("Further, …") seems to be a general statement about the

dependence of the rate of convergence on the initial geometric standard deviation.

Answer: The second sentence was confusing. As both sentences shared the same references, the one referring to the geometric standard deviation has been deleted, and the remaining sentence has been properly referenced.

• L 36: The abbreviation DSD has already been introduced in L 21.

Answer: The second definition of DSD has been deleted.

• L 44: "need to calculate a huge amount of equations, which number ranges from several dozens

to hundreds, at each grid point and time step" –> "need to calculate dozens to hundreds of

equations at each grid point and time step"

Answer: Fixed.

• L 44: Also mention numerical diffusion as one of the major problems with bin microphysics?

See e.g. [1]

Answer: While it is true that one of the major problems with bin microphysics, and microphysical calculations in general is the numerical diffusion, it is highly dependent of the numerical method used to solve the KCE. For example, the method used (Bott, 1998) is specifically designed to be mass- conservative and to limit the natural diffusiveness of the problem at hand. However, an explanation on this matter is included in the revised version of the manuscript.

• L 57: "20 μm and 41 μm being" instead of "being 20 μm and 41 μm" – I won't continue to do

"micro-corrections" of grammar and typos, but the manuscript really needs some

thorough editing for clarity and correctness (see my first general comment). Not being a

native English speaker myself, I do understand the difficulty of writing in a foreign

language, but putting some effort into this will result in a more reader-friendly paper that

stands a better chance of getting read and cited by other scientists.

Answer: Fixed.

• L 91: This introduction to machine learning seems kind of out of place, especially after the

previous paragraph already talks about deep neural networks.

Answer: The authors agree with the referee, and the paragraphs have been switched to provide more clarity for the reader.

• General remark about equations: Please define all variables involved, even if their meaning

seems straightforward – e.g., in Eq. (1), say that r is radius, in Eq. (2), say what N is, etc.

Answer: Fixed.

• Neural network architecture: How did you come up with this specific architecture? Did you try

other (e.g., simpler) architectures as well?

Answer: Initially we tried a conventional feed-forward network, very similar to the one used in (Alfonso & Zamora, 2021), which is simpler and the training process is a lot faster. The results with that architecture were good. Taking that as a base, we move forward to try different types of neural network architectures, and we learned about the cascade-forward architecture. We decided to test it and select the one with the best results. Using cascade-forward networks was a time-consuming task, but worth it in the end, as the results improved in accuracy in at least two orders of magnitude using the same number of neurons.

• L 171: The commonly used terminology in machine learning is that the training data are the

data used to fit the model, the validation data are used for model selection (e.g., when you are

testing different neural network architectures, or comparing, say, the neural network with a

random forest model, you decide on a final model based on the models' performances on the

validation data), and the test set is used for assessment of the generalization error of the final

chosen model (see e.g. [2]). Since no model selection is done in this study, what is the called

"validation set" should more appropriately be called the test set here.

Answer: The validation set have been renamed test set in the manuscript.

• L 214: How were the ranges of the $\mu$ and $\sigma$ parameters (rightmost column of Table 1) for the

uniformly random sampling of the distribution parameters that was used to generate the training

data determined? Were they "reverse engineered" based on a certain range of LWC values that

are thought to be physically reasonable?

Answer: The ranges were determined partially based on data from the CRYSTAL-FACE experiment mentioned in (Alfonso & Zamora, 2021). From that point onwards, we extended the ranges in order to cover a very extensive parameter space complementing the ranges with data from previous simulations using the original parameterization.

• L 234: I think it would be interesting to include the collision-coalescence parameterization using the trapezoidal rule to solve Eq. (13) in the results (e.g., in Figure 8) – presumably the main advantage of predicting the moment tendencies using the DNN rather than computing them using the trapezoidal rule is computational efficiency, so it would be nice to know how much faster the DNN is, as well as to see how the mass density spectra obtained using this "trapezoidal parameterized model" compare to those shown in Figure 8 (reference solution and

predicted parameterized model). See also my second general comment. Based on the good agreement between the DNN predictions and the validation targets computed using the trapezoidal rule (Figure 7), the resulting mass density spectra will probably look very similar, but I think it would still be interesting for the reader to see that comparison.

Answer: As it is correctly though by the reviewer, the results of the original parameterization and the ML-based model are similar enough not to be included in the manuscript, to avoid repetition. The main advantage that offers the use of ML is the simplification of the procedures to solve eq. 13, which is very complex to solve numerically, with the exception of using very costly numerical schemes. For instance, the standard quadrature does not apply to eq. 13, and the use of lookup tables is not among the best solutions to the problem.

• L 336: I think it's a bit of a stretch to say that the third mode in the evolution of the KCE generated spectra "is reproduced by the parameterization as a wider second mode" – it seems to

me that the parameterization is not able to capture that development.

Answer: The phrasing has been changed to reflect that fact.

**Figures**

• 7: The x axis label ("Actual Total Moment Tendencies") of M0 and M1 are missing

Answer: As M0, M1, M4 and M5 share the same x-axis label, it was omitted in M0 and M1 to avoid an overload.

**Overview of the revised manuscript:**

A general revision of the draft article was performed, including changes in its content, resulting in a slightly longer, more comprehensible draft. A summary of the main changes is included as follows:

- The Introduction has been restructured, to provide more clarity about the state of the art and a better understanding of the main ideas of the article.
- A complete grammar review of the article has been done, resulting in a paper more friendly to the reader.
- Figure 4 has been discarded, as it duplicated information from Table 1, and remaining figures have been renumbered accordingly.
- An extra parameterization has been added (WDM6), in order to compare its results with those of the Machine Learning parameterization and the explicit model. The code for this new model has been properly referenced and included in the Code Availability section of the manuscript.
- Several figures have been modified to reflect the inclusion of the additional parameterized model.
- The conclusions are now supported with the analysis of the comparisons of three models, instead of two.

**References**

Alfonso, L., & Zamora, J. M. (2021). A two-moment machine learning parameterization of the autoconversion process. *Atmospheric Research*, *249*, 105269. https://doi.org/10.1016/j.atmosres.2020.105269

Bott, A. (1998). A flux method for the numerical solution of the stochastic collection equation. *Journal of the Atmospheric Sciences*, *55*(13), 2284–2293. https://doi.org/10.1175/1520-0469(1998)055<2284:AFMFTN>2.0.CO;2

Clark, T. L. (1976). Use of log-normal distributions for numerical calculations of condensation and collection. *Journal of the Atmospheric Sciences*, *33*(5), 810–821. https://doi.org/10.1175/1520-0469(1976)033<0810:UOLNDF>2.0.CO;2

Cohard, J.-M., & Pinty, J.-P. (2000). A comprehensive two-moment warm microphysical bulk scheme. I: Description and tests. *Quarterly Journal of the Royal Meteorological Society*, *126*(566), 1815–1842. https://doi.org/10.1256/smsqj.56613

Seifert, A., & Beheng, K. D. (2001). A double-moment parameterization for simulating autoconversion, accretion and selfcollection. *Atmospheric Research*, *59–60*, 265–281. https://doi.org/10.1016/S0169-8095(01)00126-0

Swift, D. L., & Friedlander, S. . (1964). The coagulation of hydrosols by brownian motion and laminar shear flow. *Journal of Colloid Science*, *19*(7), 621–647. https://doi.org/10.1016/0095-8522(64)90085-6

---

## Author Response (AR2)

The authors thank the anonymous referees for their helpful comments that improved the quality of the manuscript.

**Comments from Anonymous Referee # 1 and answers from the authors**

*General comments*

*Overall, the paper has been significantly improved in terms of both science and writing. It is clear that the authors have taken time to take the suggestions from the reviewers into account. The additional comparison with the commonly-used parameterization P-CP2000 enables readers to objectively evaluate the predictions by the new parameterization P-DNN. The authors have clarified in their response and in the text that the case in Table 4 represents the training data well. They also emphasized that the objective of this paper is an introduction of the new parameterization and its development through machine-learning. In terms of writing, however, there are still more improvements necessary before publication so that the scientific contents can be more easily conveyed to readers. Therefore, I suggest minor revisions of the paper, mostly due to the technical corrections on writing.*

*Specific comments*

*Naming:*
*From Chapter 4 and onwards, the authors often use "parameterized/parameterization model" and "reference/explicit model/solution" to refer to P-DNN and KCE, respectively. I suggest the authors to use consistent names (e.g., P-DNN and KCE) throughout the paper so that the readers do not get confused. The authors should first introduce the naming at the beginning of Chapter 4 and then solely use it for the rest of the paper.*

Answer: The authors share the referee's concerns about naming. The manuscript has been modified to reflect this, and to be more concise and clear when referring to the different models mentioned in the paper.

*Objective of Chapter 4:*
*It is important to clarify the meaning and objectives of the experiments done in Chapters 4 and 5. Up to Chapter 3, the detailed introductions to the equations and methodology are given. In the current form of presentation, the readers may take the comparisons done in Chapters 4 and 5 as an overall evaluation of the new parameterization P-DNN, which is solely based on one representative case. However, in reality, the comparison simply serves as an experiment on how P-DNN predicts the drop size distributions on an example case – the overall evaluation of PDNN is already done on Table 3 in Chapter 3 rather than in Chapters 4 and 5. While I understand it now, I thought the comparisons in Chapters 4 and 5 served to evaluate P-DNN, and this is why I questioned why there was only one case study in the previous round of review. Therefore, I suggest that the authors add a few sentences or a paragraph at the beginning of Chapter 4 on why this experiment/comparison is done. This will allow the readers to interpret the results just as an example, rather than misunderstand that the results (e.g., within 10 % difference from KCE) apply to all the simulations by P-DNN with any initial conditions.*

Answer: The authors acknowledge that this comment is very important to the clarity of the manuscript. Thus, an introduction has been added to Chapter 4, clarifying the interpretation and objective of the experiments, to reflect that under no circumstance the experiment should be interpreted as an overall evaluation of the developed parameterization, as this has been already done in Chapter 3, but merely an illustrating example of a simulation case.

*Structure of Chapter 4:*
*If there is only one subchapter 4.1, it can be written as a paragraph in 4.*

Answer: Following the addition of an explanatory introduction as result of the previous comment, the structure of Chapter 4 changed to reflect this. It now have sections 4.1 (experiment design and initial conditions) and 4.2 (WDM6 parameterization).

*Figure 4:*
*If the authors can add one colored dot in each of these panel plots to show where the experiment case in Table 4 stands in this figure, that would be helpful for the readers to recognize how representative that case is.*

Answer: Good point. Figure 4 has been modified to add these dots. It now looks like this:

[Figure]

Also, the caption also reflects what those red dots are.

*Figure 7:*
*The simulation result with P-CP2000 is clearly different from the others, but why is it showing the existence of droplets that are smaller than the initial distribution (i.e., $r < 6$ um)? Is it because the size distribution was not physically calculated but was simply diagnosed by the prognosed distribution parameters? Please add some explanation.*

Answer: This is caused by the fixed distribution parameters employed in its formulation. The slope parameter of the gamma distribution is determined by an analytical expression and evolves with time within certain limits, but the parameters related to the spectral breadth (width) are held fixed, thus the weird width of P-CP2000's initial spectrum. I am happy to answer any question related to P-CP2000, and also additional and more detailed information about the model can be found in Cohard and Pinty (2000).

**Technical corrections**

*\*Make sure that the references are out of the brackets when necessary (e.g., line 60).*
*\*Future and present tenses are mixed, please check the consistency.*
*\*In addition to the suggestions below, I highly suggest the thorough check on the overall writing*
*by the authors.*

Answer: The authors did a complete review of the manuscript, and fixed all issues related to references in brackets (and not in brackets), and the tense used throughout the paper. Also, all specific technical suggestions have been fixed, along with some others, generally improving the clarity of the manuscript.

**Comments from Anonymous Referee # 2 and answers from the authors**

*While I do see and appreciate the effort that the authors put into revising the manuscript (namely, into the addition of the WDM6 parameterization for comparison, the restructuring of theintroduction, and the addressing of many minor reviewer comments), the major concerns that constitute my reasons for rejection are still:*

*1) Advantage of using a machine learning approach The advantage / added value of using a machine learning model for the task of predicting the moment tendencies (compared to e.g. a numerical quadrature) is not demonstrated. I am not sayingthat this value doesn't exist, but that it isn't established in the paper, and I think that it would have been important to do so. In their response, the authors write that the utility of the machine learningparameterization lies in a "more straightforward way of computing the moment tendencies", and that "the numerical solution of eq. (13) is a complex task, particularly the selection andimplementation of an efficient numerical methodor quadrature for the solution of double integrals". However, it seems that this complex task has been solved successfully by the authors, as they use a quadrature method to generate the output(target values) used to train the model. Given the demonstrated feasibility of numerically integrating the double integrals, I think that*

*simply referring to this as "a complex task" and advertising themachine learning approach as "a more straightforward way" of computing (or more precisely, predicting) the moment tendencies is not sufficient. It is probably the case that the machine learningapproach is faster than the quadrature, and that would be a good argument for why its use is advantageous, but this argument would have to be established quantitatively. I did not mean tosuggest that "an exhaustive computational or hardware-focused analysis of the problem" should be done, but if the neural network lowers the computational cost in a significant way, it should bepossible to illustrate this somehow (even if of course the exact numbers and details will depend on computational architecture), e.g. with runtime measurements for an example simulation and/orsome back-of-the-envelope calculations (where computational gains from parallelizing the predictions of the neural network could be factored in linearly). Whatever the main advantage of the neural network presented here is compared to the "competitor approaches" (it doesn't have to be computational efficiency - maybe the neural network is moreeasily adaptable to different kernels than a quadrature method?), it should be motivated / explained / demonstrated in order to make it clear that it is not a case of "machine learning for the sake of machine learning".*

Answer:

Neural networks give us a better way to estimate the values of the integral (13) in the manuscript. If the parameterization was implemented from real time calculations of the integral (13) by the trapezoidal rule every time it was necessary, it would be extremely slow. The neural networks of course do not replace the computation of integrals, but since they have the ability to learn and model complex non-linear functions, they allow us (once trained) to estimate them efficiently for values of the parameters ($N_1$, $\mu_1$, $\sigma_1$, $N_2$, $\mu_2$ and $\sigma_2$), for which it has not been previously calculated.

Before the widespread adoption of machine learning, the alternative previously used by other authors (Clark, 1976; Clark and Hall, 1983; Feingold et al., 1998) were the lookup tables, that are tables that stores a list of predefined values (the moment tendencies in this case). Then, in the context of our work, the lookup table is a mapping function that relates the parameters of the basis functions ($N_1$, $\mu_1$, $\sigma_1$, $N_2$, $\mu_2$ and $\sigma_2$), with the total moment tendencies $\left(\frac{dN\overline{R^p}}{dt}\right)$.

However, usually, functions computed from lookup tables have a limited domain. For larger problems, the memory and the time required to access the data increase substantially. Furthermore, preferably, we need functions whose domain is a set with contiguous values. Additionally, every time we need to calculate the integral (13), a search algorithm must be executed in order to retrieve the moment tendency for a given set of parameters, and some kind of interpolation will be needed to compute moment tendencies for values of the parameters for which it has not been calculated.

The advantage of the neural networks is that all the computational effort is dedicated to the training phase. Once we trained the networks, they replace the lookup tables and are able to map efficiently the parameters of the basis functions with total moment tendencies. A significant speed up is expected since we just need to evaluate the input parameters, and there is no need to execute a searching algorithm in order to retrieve the desired information.

This explanation has been added at the end of Chapter 3, with the objective of providing more clarity about this topic.

*2) Clarity of Language / Style / Grammar: While the linguistic clarity and correctness of the revised manuscript are clearly improved compared to the previous version, there are still numerous errors (grammar and word choice) and sentences that lack in clarity.*

Answer: The authors agree with the referee that a review of the entire manuscript was in order. This has been done, with the comments of both referees as guide, and the clarity of the paper has improved. Next you will find a couple of comments that we believed worth it:

- *L69 "simulates the explicit approach" –> unclear what this means*

  Answer: What we tried to say with this is that the parameterization developed by Clark (1976) and Clark and Hall (1983) differs from the traditional bulk methods in the variables it calculates. The bulk methods usually follow the evolution of one, two or more recently three selected moments of a distribution function (usually gamma), while Clark's approach follows the evolution of the parameters of those distributions (lognormals), and not the actual values of the moments per se. Thus, it "simulates" the bin approach in the way that the DSD can be easily reproduced from those parameters. Therefore, Clark follows the DSD through the explicit calculation of the distributions' parameters, and not the moments.

  Of course, the formulation of the rates of the moments is included in the system of equations, but this is incidental, and needed to close the system, not because the calculation of the moments is the objective of the parameterization. That is why we called the Clark's parameterization a "hybrid approach" to modelling cloud microphysics.

- *L 94 The abbreviation "ML" has not been defined*

  Answer: True. The ML abbreviation has been defined in the first mention of Machine Learning in the introduction.

*3) Evaluation of the new parameterization for several different experiments. This was also suggested by the other reviewer but considered outside of the objective / scope of the study by the authors. That is fair, but I still think it would have made the study more solid.*

Answer:

We believe that clarity is responsibility of the authors, since the following was not clearly stated in the manuscript or in the previous answers. It is important to clarify the meaning and objectives of the experiments done in Chapter 3 and 4. The overall evaluation of the novel components (ML approach) is done in Chapter 3, through the information in Table 3 and in Figures 5 and 6, and their related explanations and analyses in the text. The comparison introduced in Chapter 4, and results showed in Chapter 5, simply serve as an illustrating experiment on how the developed model predicts the DSD and bulk variables, on an example basis. Under no circumstance the experiments from Chapters 4 and 5 should be interpreted as an overall evaluation of P-DNN. An introduction has been added to Chapter 4 clearly explaining all this. That is why the authors considered the realization of more experiments outside of the scope of the manuscript.

**References**

Clark, T. L.: Use of Log-Normal Distributions for Numerical Calculations of Condensation and Collection, J. Atmos. Sci., 33(5), 810–821, doi:10.1175/1520-0469(1976)033<0810:UOLNDF>2.0.CO;2, 1976.

Clark, T. L. and Hall, W. D.: A Cloud Physical Parameterization Method Using Movable Basis Functions: Stochastic Coalescence Parcel Calculations, J. Atmos. Sci., 40(7), 1709–1728, doi:10.1175/1520-0469(1983)040<1709:ACPPMU>2.0.CO;2, 1983.

Cohard, J.-M. and Pinty, J.-P.: A comprehensive two-moment warm microphysical bulk scheme. I: Description and tests, Q. J. R. Meteorol. Soc., 126(566), 1815–1842, doi:10.1256/smsqj.56613, 2000.

Feingold, G., Walko, R. L., Stevens, B. and Cotton, W. R.: Simulations of marine stratocumulus using a new microphysical parameterization scheme, Atmos. Res., 47–48, 505–528, doi:10.1016/S0169-8095(98)00058-1, 1998.

---

## Author Response (AR3)

The authors thank the editor for his helpful comments that improved the quality of the manuscript.

**Comments from editor and answers from the authors**

*General comments*

*In general, in my opinion, the discussion of figures 9 and 10 and the conclusions section are the weakest points of the manuscript as of now. I suggest a rewrite of the relevant paragraphs, and provide further suggestions below.*

Answer: The discussion of Figures 9 and 10 have been improved, including now arguments related to the physical consistency of the variables depicted.

*Moreover, as underlined earlier by one of the reviewers, any conclusions referring to the computational cost have no support in the presented analysis. Please thus preferably cover computational cost in the analysis, or alternatively refrain from stating that the introduced approach offers improvement in this regard. Several mentions of the expensive look-up tables should best be replaced with a quantitative analysis of their cost, or removed.*

Answer: The relevant parts of the manuscript have been changed to refrain from stating improvements about computational efficiency of the developed model or lookup tables.

*As I have indicated in the very first message regarding this submission, I suggest mentioning the breakup process. It is listed as one of the missing mechanisms in the work of Clark (1976). Moreover, it is included in the Cohard & Pinty (2000) formulation (p. 1826 therein) to which the comparison is made, while it is not part of the Bott reference solution.*

Answer: The objective of the research was to reproduce the behavior of the parameterization developed by Clark (1976) and Clark and Hall (1983) via a Machine Learning approach. Since the original parameterization does not include drop breakup in its formulation, it has been left out of the current implementation as well. However, the manuscript now includes mentions to this matter, and the inclusion of the breakup process has been included as one important recommendation for future versions of the model.

*To fulfill the archival requirements of GMD, a persistent archive for the coad1d.f file is required (and corresponding change the code availability section). Personal university profile websites are not considered as permanent archives.*

Answer: We have contacted Andreas Bott again, and with his permission we have stored the coad1d.f file in Zenodo. It now can be found at https://doi.org/10.5281/zenodo.5660185. The code availability section has been updated accordingly.

**Specific comments**

All specific comments have been addressed and fixed. Here we only mention those that require further explanation.

*p3/l66: "is very expensive computationally" is too vague, and in fact misleading given that particle-based approaches are being introduced as less computationally expensive than the bin schemes covered in the preceding paragraph, please elaborate and refer to literature (perhaps Morrison et al. 2020: https://doi.org/10.1029/2019MS001689)*

Answer: The Lagrangian particle-based is accurate, and represents well the stochastic nature of the collision-coalescence of drops, but it is also computationally expensive, as a large number of particles are needed in each grid cell, to be able to calculate accurate statistics (Morrison et al., 2020). The cost of these schemes could be reduced by using simple methods to treat droplet activation, such as the Twomey CNN activation (Grabowski et al., 2018; Twomey, 1959). However, even considering those simplifications, the cost of a Lagrangian particle-based scheme is 25% greater than bin microphysics, when considering a similar number of particles and bin variables per grid cell (Grabowski, 2020). This argument has been added to the introduction in order to elaborate on this topic.

*p6/l154,156: bold notation for F*

Answer: F from equations 13 and 14 are not the same vector F from eq 7. However, to gain in clarity, F from eqs 13 and 14 has been relabeled as B.

*p7/l169: x_c seem undefined, wouldn't a reference to eq. (12) be enough anyway?*

Answer: Absolutely true. Equation 16 has been removed from the manuscript, and replaced with a reference to eq. 12. Also, variable $x$ has been replaced with $m$ to make the notation consistent, and a definition of $m_c$ has been added to the corresponding paragraph.

*p7/l181: "and can be used with authorization of the author" is puzzling as the code is publicly accessible, please establish proper licensing and versioning terms with the author and cover it in the code availability section (not in the text as is done now)*

Answer: The code have been versioned (v1.0.0) and licensed (GNU Affero General Public License v3.0 or later). It can now be found at https://doi.org/10.5281/zenodo.5660185. The code availability section has been updated accordingly.

*p15/l302-303: this sentence seems unneeded given the "values require normalization"*

*statement on page 12*

Answer: This sentence was added in a previous review iteration because one of the referees mentioned that the axles did not match and that I should explain why. Since it is the result of previous reviews, we consider that it should not be changed in this iteration.

*p18/Fig 7: y axis unit: "lnr-1" -> "ln(r/1 m)-1", right?*

Answer: As the mass density function $g$ is defined in function of $\ln r$, the plots are in units of $\ln r$ too. An example of this units ($\ln r^{-1}$) being used in previous literature can be found at Berry (1967).

*p20/Fig 8: y unit wrong? if it is a number density, then the x axis unit should be featured for the area-under-the-curve to sum up to N in cm-3*

Answer: The units for number concentration DSD are *number of drops per unit volume* $(cm^{-3})$ in the radius range $(r, r + dr)$. To calculate the concentration probability density function, lognormal distributions need to be integrated over radius (Pruppacher and Klett, 2010)

$$N_i = \int_{r_{i-1}}^{r_i} f(r_i)dr$$

with *f(r)* being the lognormal probability density function as defined in eq. 2 of the manuscript:

$$f(r) = \frac{N}{\sqrt{2\pi}\sigma r} e^{[-(\ln r - \mu)^2/(2\sigma^2)]}$$

Of course, during the course of the calculations all units must the homogeneous (*cm*). The *x* axis in Fig. 8 is depicted in *μm* for clarity, to avoid the space-consuming scientific notation needed to express it in *cm*.

*p22/l434: which observations? give reference, elaborate*

Answer: Barros et al. (2008) found the same behavior while revisiting the validity of the experimental results obtained by Low and List (1982), excluding drop breakup. This piece of information has been added to the manuscript to provide clarity.

**References**

Barros, A. P., Prat, O. P., Shrestha, P., Testik, F. Y. and Bliven, L. F.: Revisiting Low and List (1982): Evaluation of raindrop collision parameterizations using laboratory observations and modeling, J. Atmos. Sci., 65(9), 2983–2993, doi:10.1175/2008JAS2630.1, 2008.

Berry, E. X.: Cloud droplet growth by collection, J. Atmos. Sci., 24(6), 688–701, doi:10.1175/1520-0469(1967)024<0688:CDGBC>2.0.CO;2, 1967.

Clark, T. L.: Use of log-normal distributions for numerical calculations of condensation and collection, J. Atmos. Sci., 33(5), 810–821, doi:10.1175/1520-0469(1976)033<0810:UOLNDF>2.0.CO;2, 1976.

Clark, T. L. and Hall, W. D.: A cloud physical parameterization method using movable basis functions: Stochastic coalescence parcel calculations, J. Atmos. Sci., 40(7), 1709–1728, doi:10.1175/1520-0469(1983)040<1709:ACPPMU>2.0.CO;2, 1983.

Grabowski, W. W.: Comparison of Eulerian bin and Lagrangian particle-based schemes in simulations of Pi Chamber dynamics and microphysics, J. Atmos. Sci., 77(3), 1151–1165, doi:10.1175/JAS-D-19-0216.1, 2020.

Grabowski, W. W., Dziekan, P. and Pawlowska, H.: Lagrangian condensation microphysics with Twomey CCN activation, Geosci. Model Dev., 11(1), 103–120, doi:10.5194/gmd-11-103-2018, 2018.

Low, T. B. and List, R.: Collision, coalescence and breakup of raindrops. Part I: Experimentally established coalescence efficiencies and fragment size distributions in breakup, J. Atmos. Sci., 39(7), 1591–1606, doi:10.1175/1520-0469(1982)039<1591:CCABOR>2.0.CO;2, 1982.

Morrison, H., Lier-Walqui, M., Fridlind, A. M., Grabowski, W. W., Harrington, J. Y., Hoose, C., Korolev, A., Kumjian, M. R., Milbrandt, J. A., Pawlowska, H., Posselt, D. J., Prat, O. P., Reimel, K. J., Shima, S., Diedenhoven, B. and Xue, L.: Confronting the challenge of modeling cloud and precipitation microphysics, J. Adv. Model. Earth Syst., 12(8), e2019MS001689, doi:10.1029/2019MS001689, 2020.

Pruppacher, H. R. and Klett, J. D.: Microphysics of clouds and precipitation, Springer Netherlands, Dordrecht., 2010.

Twomey, S.: The nuclei of natural cloud formation part II: The supersaturation in natural clouds and the variation of cloud droplet concentration, Geofis. Pura e Appl., 43(1), 243–249, doi:10.1007/BF01993560, 1959.

---

## Author Response (AR4)

The authors thank the editor for his helpful comments that improved the quality of the manuscript

**Comments from editor and answers from the authors**

All comments have been addressed and fixed. Here we only mention those that require further explanation.

*The figure archive was uploaded apparently as the electronic supplement file (please make sure that the editorial office is informed that it is not meant to be published as electronic supplement).*

Answer: We will make sure that the editorial office is informed about this via email. Thanks for noticing this.

*Page 1 / line 21-22: suggest rephrasing the reference to Marshall and Palmer - certainly this is a seminal work, yet not a "first attempt at characterizing drop spectra" (cf., e.g., Houghton 1932, "The Size and Size Distribution of Fog Particles", https://doi.org/10.1063/1.1745072; Schumann 1940, "Theoretical aspects of the size distribution of fog particles", https://doi.org/10.1002/qj.49706628508). Please clarify in which aspects Marshall and Palmer's work was pioneering.*

Answer: Since that is one of the opening statements, a brief explanation have been added about the pioneering aspects of the work of Marshall and Palmer (1948). Marshall and Palmer (1948) were apparently the first to describe the size distributions of raindrops in space, as opposed to those distributions over a surface. The surface distributions of precipitation particle sizes are still of interest and information about the spatial distributions can be computed from surface data. Marshall and Palmer, however, were concerned with the size distributions of the drops in the atmosphere aloft where they could be viewed by radar (Smith, 1982).

*Page 1 / line 26-27: not even a suggestion, just sharing a recent find: Andersson 2021, "Mechanisms for log normal concentration distributions in the environment", https://doi.org/10.1038/s41598-021-96010-6*

Answer: Interesting work, we will definitely check it out. Thank you!

*Page 2 / lines 68-71: let me again raise the issue of referring to particle-resolved methods*

*as more costly than bin. The referenced work of Grabowski (2020) (i) is a condensation-only study,*

*(ii) it employs a serial implementation of the particle-based microphysics and (iii) it focuses on a small-scale cloud-chamber experiment setup. It is thus not representative for at least these three*

*reasons. At this point, I suggest to remove the discussion of computational cost of particle-*

*based methods from the paper, as it is not directly relevant to presented material. [If intending to cover it however, please elaborate why, even for single attribute, spatial transport of particles (trivial ODEs) would be costlier than solving transport of bin-microphysics scalar fields (advection PDEs); why a Monte-Carlo coalescence algorithm with linear scaling would.*

Answer: We eliminated the discussion related to computational cost of particle-based methods, since it is not relevant to the manuscript.

*Page 2 / line 38-40: "the KCE has no analytical solution" - it has for some kernels; "numerical schemes, which are very diffusive by nature" - in atmospheric modelling context likely so, but in principle it depends on the grid choice, right? Suggest rephrasing.*

Answer: Indeed, it depends on the grid choice and numerical method used to solve it. This has been noted in the manuscript.

*Page 3 / line 81: I admit, I don't understand the "However, this integration can be made only once*

*for all parameters at each time step". Suggest clarifying.*

Answer: The sentence was misleading and prone to confusion. Also, it did not contribute much to the explanation of the parameterization. Thus, it has been eliminated from the manuscript.

*Page 10 / Table 1: please clarify that \mu is given in natural logarithm of metres, right? (it would be clearer to give exp(\nu) geometric mean values in micrometres)*

 *Page 16 / Table 4: ditto*

Answer: The units of $\mu$ has been added to the manuscript. The units are $\ln cm$, since the lognormal distribution works in $cm$. Also, as $\mu$ is a parameter of the distribution function, we prefer to keep it in units of $\ln cm$, to avoid confusion about which values where actually used in the model.

*Page 20 / Figure 8: I read carefully your answer, yet it only assures me that (as in the case of Fig. 7, the vertical axis unit label should include "ln(r^{-1})" or alike. Note that in aerosol studies, base-10 logarithms are commonly used, and it is perhaps worth to remind readers of the logarithm base as well.*

Answer: Figure 8 y-label has been changed accordingly.

**References**

Marshall, J. S. and Palmer, W. M. K.: The distribution of raindrops with size, J. Meteorol., 5(4), 165–166, doi:10.1175/1520-0469(1948)005<0165:TDORWS>2.0.CO;2, 1948.

Smith, P. L.: On the graphical presentation of raindrop size data, Atmosphere-Ocean, 20(1), 4–16, doi:10.1080/07055900.1982.9649124, 1982.